# Role of jellyfish in the plankton ecosystem revealed using a global ocean biogeochemical model

Rebecca M. Wright[1, 2], Corinne Le Quéré[1], Erik Buitenhuis[1], Sophie Pitois[2], Mark Gibbons[3]

[1]Tyndall Centre for Climate Change Research, School of Environmental Sciences, University of East Anglia, Norwich, NR4 7TJ, UK

[2]Centre for Environment, Fisheries & Aquaculture Science, Lowestoft, NR33 0HT, UK

[3]Department of Biodiversity and Conservation Biology, University of the Western Cape, Cape Town, Bellville 7535, RSA

*Correspondence to*: Rebecca M. Wright (rebecca.wright@uea.ac.uk)

**Abstract.** Jellyfish are increasingly recognised as important components of the marine ecosystem, yet their specific role is poorly defined compared to that of other zooplankton groups. This paper presents the first global ocean biogeochemical model that includes an explicit representation of jellyfish and uses the model to gain insight into the influence of jellyfish on the plankton community. The PlankTOM11 model groups organisms into Plankton Functional Types (PFT). The jellyfish PFT is parameterised here based on our synthesis of observations on jellyfish growth, grazing, respiration and mortality rates as functions of temperature and on jellyfish biomass. The distribution of jellyfish is unique compared to that of other PFTs in the model. The jellyfish global biomass of 0.13 PgC is within the observational range, and comparable to the biomass of other zooplankton and phytoplankton PFTs. The introduction of jellyfish in the model has a large direct influence on the crustacean macrozooplankton PFT and influences indirectly the rest of the plankton ecosystem through trophic cascades. The zooplankton community in PlankTOM11 is highly sensitive to the jellyfish mortality rate, with jellyfish increasingly dominating the zooplankton community as its mortality diminishes. Overall, the results suggest that jellyfish play an important role in regulating global marine plankton ecosystems across plankton community structure, spatiotemporal dynamics, and biomass, a role which has been generally neglected so far.

# 1 INTRODUCTION

Gelatinous zooplankton are increasingly recognised as influential organisms in the marine environment, not just for the disruptions they can cause to coastal economies (fisheries, aquaculture, beach closures and power plants etc.; Purcell et al., 2007), but also as important consumers of plankton (Lucas and Dawson, 2014), a food source for many marine species (Lamb et al., 2017) and as key components in marine biogeochemical cycles (Crum et al., 2014; Lebrato et al., 2012). The term gelatinous zooplankton can encompass a wide range of organisms across three phyla: Tunicata (salps), Ctenophora (comb-jellies), and Cnidaria (true jellyfish). This study focuses on Cnidaria (including Hydrozoa, Cubozoa and Scyphozoa), which contribute 92% of the total global biomass of gelatinous zooplankton (Lucas et al., 2014). The other gelatinous zooplankton groups, Tunicata and Ctenophora, are excluded from this study because there is far less data available on their biomass and vital rates than for Cnidaria, and they only contribute a combined global biomass of 8% of total gelatinous zooplankton (Lucas et al., 2014). Cnidaria are both independent enough from other gelatinous zooplankton, and cohesive enough to be represented as a single Plankton Functional Type (PFT) for global modelling (Le Quéré et al., 2005). For the rest of this paper pelagic Cnidaria are referred to as jellyfish.

Jellyfish exhibit a radially symmetrical body plan and are characterised by a bell-shaped body (medusae). Swimming is achieved by muscular, "pulsing" contractions and animals have one opening for both feeding and excretion. Most scyphozoans and cubozoans, and many hydrozoans, follow a meroplanktonic life cycle. A sessile (generally) benthic polyp buds off planktonic ephyrae asexually. These, in turn, grow into medusae that reproduce sexually to generate planula larvae, which then settle and transform into polyps. Within this general life cycle, there is large reproductive and life cycle variety, including some holoplanktonic species that skip the benthic polyp stage as well as holobenthic species that skip the pelagic phase, and much plasticity (Boero et al., 2008; Lucas and Dawson, 2014).

Jellyfish are significant consumers of plankton, feeding mostly on zooplankton using tentacles and/or oral arms containing stinging cells called nematocysts (Lucas and Dawson, 2014). The large body size to carbon content ratio of jellyfish creates a low maintenance, large feeding structure, which, because they do not use sight to capture prey, allow them to efficiently clear plankton throughout 24 hours (Acuña et al., 2011; Lucas and Dawson, 2014). Jellyfish are connected to lower trophic levels, with the ability to influence the plankton ecosystem structure and thus the larger marine ecosystem through trophic cascades (Pitt et al., 2007, 2009; West et al., 2009). Jellyfish have the ability to rapidly form large high-density aggregations known as blooms that can temporarily dominate local ecosystems (Graham et al., 2001; Hamner and Dawson, 2009). Jellyfish contribute to the biogeochemical cycle through two main routes; from life through feeding processes, including the excretion of faecal pellets, mucus and messy-eating, and from death, through the sinking of carcasses (Chelsky et al., 2015; Lebrato et al., 2012, 2013a; Pitt et al., 2009). The high biomass achieved during jellyfish blooms, and the rapid sinking of excretions from feeding and carcasses from such blooms, make them a potentially significant vector for carbon export (Lebrato et al., 2013a, 2013b; Luo et al., 2020).

Anthropogenic impacts from climate change, such as increasing temperature and acidity (Rhein et al., 2013), and fishing, through the removal of predators and competitors (Doney et al., 2012), impact the plankton including

jellyfish (Boero et al., 2016; but see Richardson and Gibbons, 2008). Multiple co-occurring impacts make it
difficult to understand the role of jellyfish in the marine ecosystem, and how the role may be changed by the co-
occurring impacts. The paucity of historical jellyfish biomass data, especially outside of coastal regions and the
Northern Hemisphere, has made it difficult to establish jellyfish global spatial distribution, biomass and trends
from observations (Brotz et al., 2012; Condon et al., 2012; Gibbons and Richardson, 2013; Lucas et al., 2014; Pitt
et al., 2018).
Models are useful tools to help understand the interactions of multiple complex drivers in the environment. This
paper describes the addition of jellyfish to the PlankTOM10 global ocean biogeochemical model, which we call
PlankTOM11. PlankTOM10 represents explicitly 10 PFTs; six phytoplankton, one bacteria and three zooplankton
(Le Quéré et al., 2016). The three zooplankton groups are protozooplankton (mainly heterotrophic flagellates and
ciliates), mesozooplankton (mainly copepods) and macrozooplankton (as crustaceans, mainly euphausiids; see
Table 1 for definitions). Jellyfish is therefore the fourth zooplankton group and 11th PFT in the PlankTOM model
series. It introduces an additional trophic level to the ecosystem. To our knowledge, this is the first and only
representation of jellyfish in a global ocean biogeochemical model at the time of writing. PlankTOM11 is used to
help quantify global jellyfish biomass and the role of jellyfish for the global plankton ecosystem.
# 2 METHODS
## 2.1 PLANKTOM11 MODEL DESCRIPTION

PlankTOM11 was developed starting from the 10 PFT version of the PlankTOM model series (Le Quéré et al.,
2016), by introducing jellyfish as an additional trophic level at the top of the plankton food web (Fig. 1a). A full
description of PlankTOM10 is published in Le Quéré et al. (2016), including all equations and parameters. Here
we provide an overview of the model development, focussing on the parameterisation of the growth and loss rates
of jellyfish and how these compare to the other macrozooplankton group. We also describe the update of the
relationship used to describe the growth rate as a function of temperature and subsequent tuning. The formulation
of the growth rate is the only equation that has changed since the previous version of the model (Le Quéré et al.,
2016), although many parameters have been modified (Sect. 2.1.6).
PlankTOM11 is a global ocean biogeochemistry model that simulates plankton ecosystem processes and their
interactions with the environment through the representation of 11 PFTs (Fig. 1). The 11 PFTs consist of six
phytoplankton (picophytoplankton, nitrogen-fixing cyanobacteria, coccolithophores, mixed phytoplankton,
diatoms and *Phaeocystis*), bacteria, and four zooplankton (Table 1). Physiological parameters are fixed within
each PFT, and therefore, within-PFT diversity is not included. Spatial variability within PFTs is represented
through parameter-dependence on environmental conditions including temperature, nutrients, light and food
availability.

## (a) Plankton food web

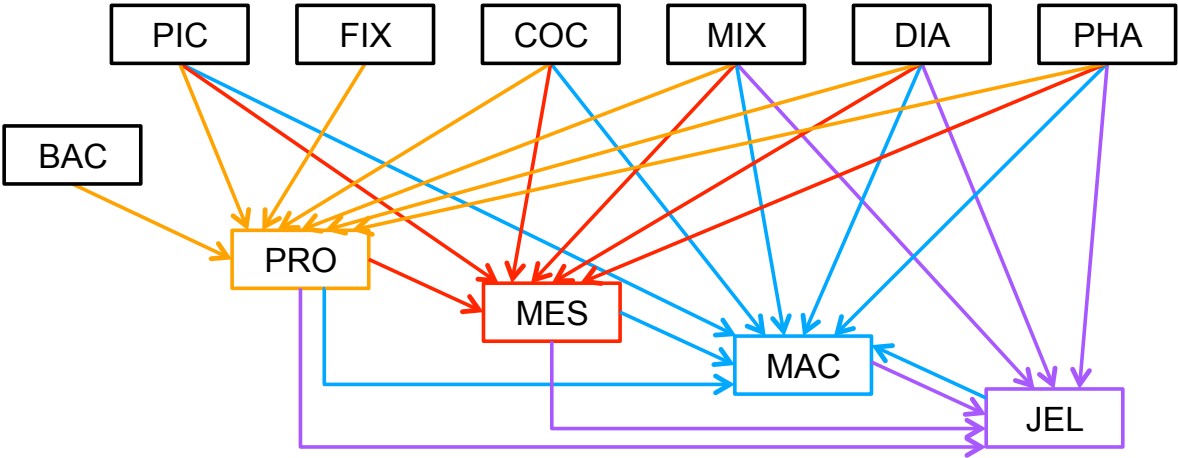

## (b) Sources and sinks for organic carbon

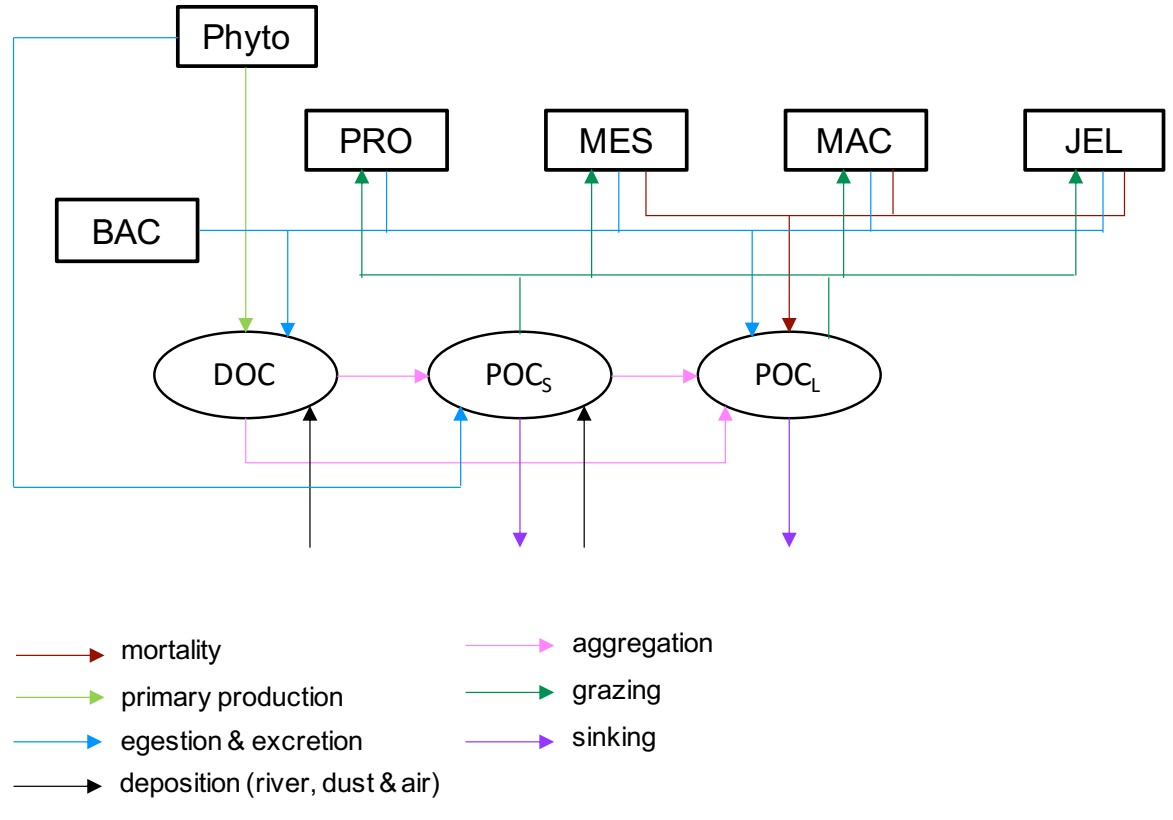

*Figure 1. Schematic representation of the PlankTOM11 marine ecosystem model (see Table 1 for PFT definitions). (a) The plankton food web, arrows represent the grazing fluxes by protozooplankton (orange), mesozooplankton (red), macrozooplankton (blue) and jellyfish zooplankton (purple). Only fluxes with relative preferences above 0.1 are shown (see Table 3). (b) Source and sinks for dissolved organic carbon (DOC) and small ($POC_S$) and large ($POC_L$) particulate organic carbon.*

The model contains 39 biogeochemical tracers, with full marine cycles of key elements carbon, oxygen, phosphorus and silicon, and simplified cycles of nitrogen and iron. There are three detrital pools: dissolved organic

carbon (DOC), small particulate organic carbon (POC$_S$) and large particulate organic carbon (POC$_L$). The elements enter through riverine fluxes and are cycled and generated through the PFTs via feeding, faecal matter, messy-eating and carcases (Fig. 1b; see Sect. 2.1.5. for detail; Buitenhuis et al., 2006, 2010, 2013a; Le Quéré et al., 2016). Model parameters are based on observations where available. A global database of PFT carbon biomass that was designed for model studies (Buitenhuis et al., 2013b) and global surface chlorophyll from satellite observations (SeaWiFS) are used to guide the model developments.

**Table 1.** Size range and descriptions of PFT groups used in PlankTOM11. Adapted from Le Quéré et al. (2016).

| Name | Abbreviation | Size Range $\mu$m | Description/Includes |
|---|---|---|---|
| **Autotrophs** | | | |
| Pico-phytoplankton | PIC | 0.5 – 2 | Pico-eukaryotes and non N$_2$-fixing cyanobacteria such as *Synechococcus* and *Prochlorococcus* |
| N$_2$-fixers | FIX | 0.7 – 2 | *Trichodesmium* and N$_2$-fixing unicellular cyanobacteria |
| Coccolithophores | COC | 5 – 10 | |
| Mixed-phytoplankton | MIX | 2 – 200 | e.g. autotrophic dinoflagellates and chrysophytes |
| Diatoms | DIA | 20 – 200 | |
| *Phaeocystis* | PHA | 120 – 360 | Colonial *Phaeocystis* |
| **Heterotrophs** | | | |
| Bacteria | BAC | 0.3 – 1 | Here used to subsume both heterotrophic *Bacteria* and *Archaea* |
| Protozooplankton | PRO | 5 – 200 | e.g. heterotrophic flagellates and ciliates |
| Mesozooplankton | MES | 200 – 2000 | Predominantly copepods |
| Macrozooplankton | MAC | >2000 | Euphausiids, amphipods, and others, known as crustacean macrozooplankton |
| Jellyfish zooplankton | JEL | 200 – >20,000 | Cnidaria medusae, 'true jellyfish' |

The PlankTOM11 marine biogeochemistry component is coupled online to the global ocean general circulation model Nucleus for European Modeling of the Ocean version 3.5 (NEMO v3.5). We used the global configuration with a horizontal resolution of 2° longitude by a mean resolution of 1.1° latitude using a tripolar orthogonal grid. The vertical resolution is 10m for the top 100m, decreasing to a resolution of 500m at 5km depth, and a total of

30 vertical z-levels (Madec, 2013). The ocean is described as a fluid using the Navier-Stokes equations and a
nonlinear equation of state (Madec, 2013). NEMO v3.5 explicitly calculates vertical mixing at all depths using a
turbulent kinetic energy model and sub-grid eddy induced mixing. The model is interactively coupled to a
thermodynamic sea-ice model (LIM version 2; Timmermann et al., 2005).
The temporal ($t$) evolution of zooplankton concentration ($Z_j$), including the jellyfish PFT, is described through
the formulation of growth and loss rates as follows:
$\frac{\partial Z_j}{\partial t} = \sum_k g_{F_k}^{Z_j} \times F_k \times MGE \times Z_j - \sum_{k=1}^{4} g_{Z_j}^{Z_k} \times Z_k \times Z_j - R_{0°}^{Z_j} \times d_{Z_j}^{T} \times Z_j$           *(1)*
*growth through grazing − loss through grazing − basal respiration*
$- m_{0°}^{Z_j} \times c_{Z_j}^{T} \times \frac{Z_j}{K_{1/2}^{Z_j} + Z_j} \times \sum_i P_i$
*− mortality*
For growth through grazing, $g_{F_k}^{Z_j}$ is the grazing rate by zooplankton $Z_j$ on food source $F_k$. This is a temperature-
dependent Michaelis-Menten term that includes grazing preference (see Sect. 2.1.2.). $MGE$ is the modelled growth
efficiency (Buitenhuis et al., 2010). For loss through grazing, $g_{Z_j}^{Z_k}$ is the grazing of other zooplankton on $Z_j$. For
basal respiration, $R_{0°}^{Z_j}$ is the respiration rate at 0°C, $T$ is temperature, $d_{Z_j}$ is the temperature dependence of
respiration ($d^{10} = Q_{10}$). Mortality is the closure term of the model and is mostly due to predation by higher trophic
levels than are represented by the model. $m_{0°}^{Z_j}$ is the mortality rate at 0°C, $c_{Z_j}$ is the temperature dependence of
the mortality ($c^{10} = Q_{10}$) and $K_{1/2}^{Z_j}$ is the half saturation constant for mortality. $\sum_i P_i$ is the sum of all PFTs,
excluding bacteria, and is used as a proxy for the biomass of predators not explicitly included in the model. More
details on each term are provided below and parameter values are given in Tables 2 through 5.

## 2.1.1 PFT Growth


Growth rate is the trait that most distinguishes PFTs in models (Buitenhuis et al., 2006, 2013a). Jellyfish growth
rates were compiled as a function of temperature from the literature (see Appendix Table A1). In previous
published versions of the PlankTOM model, growth as a function of temperature ($\mu^T$) was fitted with two
parameters:
$\mu^T = \mu_0 \times Q_{10}^{\frac{T}{10}}$           (2)
where $\mu_0$ is the growth at 0°C, $Q_{10}$ is the temperature dependence of growth derived from observations, and $T$ is
the temperature (Le Quéré et al., 2016). Jellyfish growth rate is poorly captured by an exponential fit to
temperature. To better capture the observations, the growth calculation has now been updated with a three-
parameter growth rate, which produces a bell-shaped curve centred around an optimal growth rate at a given
temperature (Fig. 2 and Table 2). The three-parameter fit is suitable for the global modelling of plankton because
it can represent an exponential increase if the data support this (Schoemann et al., 2005). The growth rate as a
function of temperature ($\mu^T$) is now defined by; the optimal temperature ($T_{opt}$), maximum growth rate ($\mu_{max}$) at
$T_{opt}$, and the temperature interval ($dT$):
$$\mu^T = \mu_{max} \times exp\left[\frac{-(T - T_{opt})^2}{dT^2}\right] \qquad (3)$$

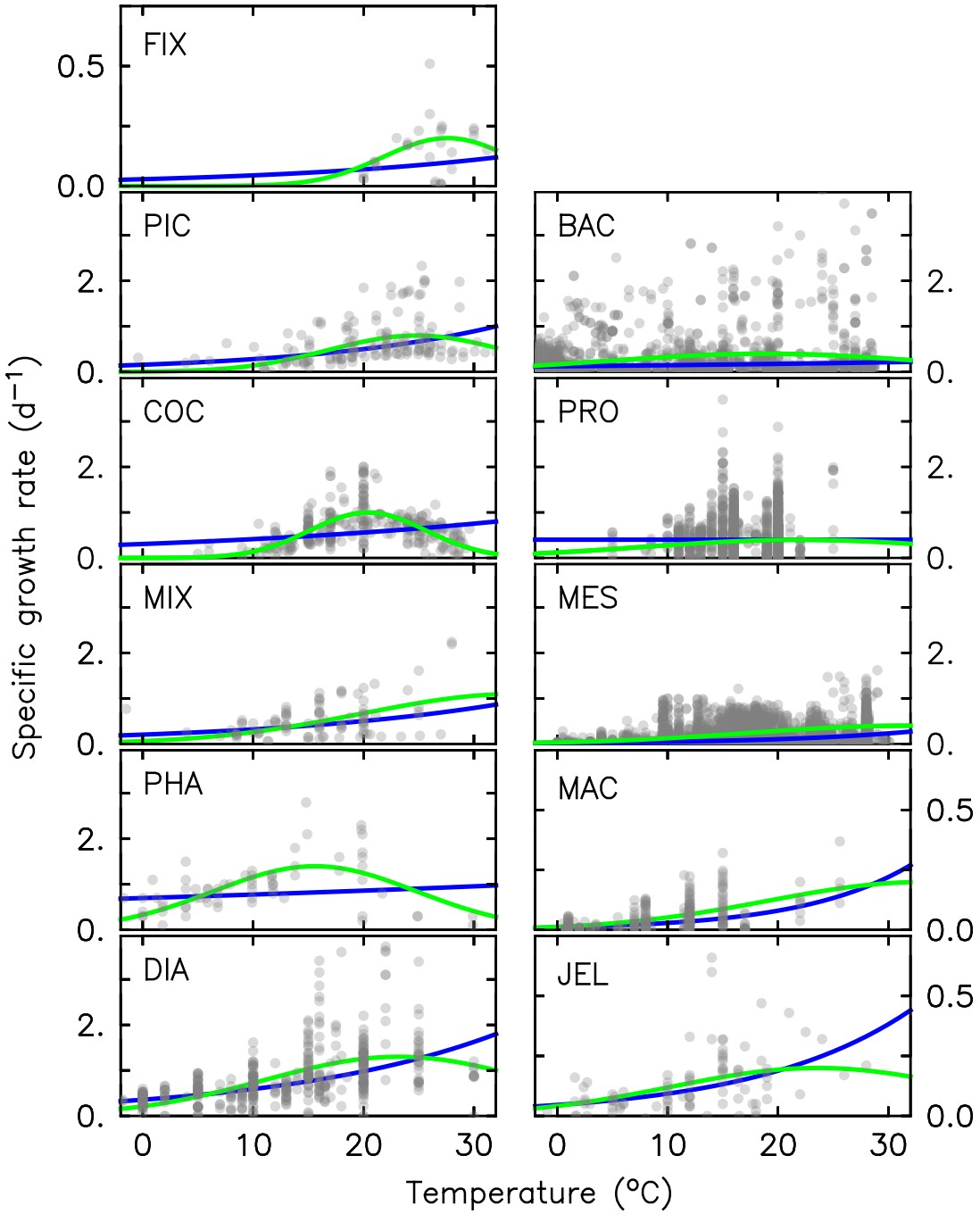


*Figure 2. Maximum growth rates for the 11 PFTs as a function of temperature from observations (grey circles). The three-*
*parameter fit to data is shown in green and the two-parameter fit is shown in blue, using the parameter values from Table 2.*
*For full PFT names see Table 1. The $R^2$ for both fits to data are given in Appendix Table A2.*

**Table 2.** Parameters used to calculate PFT specific growth rate with three-parameter fit (Eq. 3) in PlankTOM11.

| PFT | $\mu_{max}$ (d$^{-1}$) | $T_{opt}$ (°C) | dT (°C) |
|-----|------------------------|----------------|---------|
| FIX | 0.2 | 27.6 | 8.2 |
| PIC | 0.8 | 24.8 | 11.2 |
| COC | 1.0 | 20.4 | 7.4 |
| MIX | 1.1 | 34.0 | 20.0 |
| PHA | 1.4 | 15.6 | 13.0 |
| DIA | 1.3 | 23.2 | 17.2 |
| BAC | 0.4 | 18.8 | 20.0 |
| PRO | 0.4 | 22.0 | 20.0 |
| MES | 0.4 | 31.6 | 20.0 |
| MAC | 0.2 | 33.2 | 20.0 |
| JEL | 0.2 | 23.6 | 18.8 |


The available observations measure growth rate, but the model requires specification of the grazing rate (Eq. 1).
Growth of zooplankton and grazing ($g^T$) are related through the gross growth efficiency (GGE):
$$g^T = \frac{u^T}{GGE} \qquad\qquad (4)$$
GGE is the portion of grazing that is converted to biomass. This was previously collated by Moriarty (2009) from
the literature for crustacean and gelatinous macrozooplankton for the development of PlankTOM10. We extracted
data for jellyfish from this collation (all scyphomedusae) which gave an average GGE of 0.29 ± 0.27, n=126
(Moriarty, 2009).

## 2.1.2 Jellyfish PFT Grazing

The food web, and thus the trophic level of PFTs is determined through grazing preferences. The relative
preference of jellyfish zooplankton for the other PFTs was determined through a literature search (Colin et al.,
2005; Costello and Colin, 2002; Flynn and Gibbons, 2007; Malej et al., 2007; Purcell, 1992, 1997, 2003; Stoecker

et al., 1987; Uye and Shimauchi, 2005a; see Appendix Table A3 for further detail). The dominant food source was mesozooplankton (specifically copepods), followed by proto-zooplankton (most often ciliates) and then macrozooplankton (Table 3). There is little evidence in the literature for jellyfish actively consuming autotrophs. One of the few pieces of evidence is a gut content analysis where 'unidentified protists… some chlorophyll bearing' were found in a small medusa species (Colin et al., 2005). Another is a study by Boero et al. (2007) which showed that very small medusae such as *Obelia* will consume bacteria and may consume phytoplankton. Studies on the diet of the ephyrae life cycle stage are limited in comparison to those on medusa, but the literature does show evidence for ephyrae consuming protists and phytoplankton (Båmstedt et al., 2001; Morais et al., 2015). We assume that ephyrae are likely to have a higher preference for autotrophs, due to their smaller size as with the small medusa, but that this will have a minimal effect on the overall preferences and the biomass consumed, so preferences for autotrophs are kept low. Once the relative preference is established, the absolute value of the preference is tuned to improve the biomass of the different PFTs, as in Le Quéré et al. (2016). Table 3 shows the relative preference of jellyfish for its prey assigned in the model, along with the preferences of the other zooplankton PFTs. The zooplankton relative preferences are based around a predator-prey size ratio, which by design is set to 1 for zooplankton-diatom. Preferences to other PFTs and to particulate carbon are then set relative to the preference for diatoms. The preference ratios are weighted using the global carbon biomass for each type against a total food biomass weighted mean (sum of all the PFTs), calculated from the MAREDAT database, following the methodology used for the other PFTs (Buitenhuis et al., 2013a; Le Quéré et al., 2016). Zooplankton grazing is calculated using:

$$g_{F_k}^{Z_j} = \mu^T \frac{p_{F_k}^{Z_j}}{K_{1/2}^{Z_j} + \sum p_{F_k}^{Z_j} F_k} \qquad (5)$$

where $g_{F_k}^{Z_j}$ is the grazing rate by zooplankton $Z_j$ on food source $F_k$ as shown in Eq. 1, where $\mu^T$ is the growth rate of zooplankton (Eq. 3), $p_{F_k}^{Z_j}$ is the preference of the zooplankton for the food source (prey) and $K_{1/2}^{Z_j}$ is the half saturation constant of zooplankton grazing. The parameter values for grazing used in the model are given in Table 4.

### 2.1.3 Jellyfish PFT Respiration

Previous analysis of respiration rates of jellyfish found that temperature manipulation experiments with $Q_{10}$ values of >3 were flawed because the temperature was changed too rapidly (Purcell, 2009; Purcell et al., 2010). In a natural environment, jellyfish gradually acclimate to temperature changes which has a smaller effect on their respiration rates. Purcell et al. (2010) instead collated values from experiments that measured respiration at ambient temperatures, providing a range of temperature data across different studies. They found that $Q_{10}$ for respiration was 1.67 for *Aurelia* species (Purcell, 2009; Purcell et al., 2010). Moriarty (2009) collated a respiration dataset for zooplankton, including gelatinous zooplankton, using a similar selectivity as Purcell et al. (2010) for experimental temperature, feeding, time in captivity and activity levels. Jellyfish were extracted from the Moriarty

**Table 3.** Relative preference, expressed as a ratio, of zooplankton for food (grazing) used in PlankTOM11. For each zooplankton the preference ratio for diatoms is set to 1.

| PFT | PRO | MES | MAC | JEL |
|---|---|---|---|---|
| **Autotrophs** | | | | |
| FIX | 2 | 0.1 | 0.1 | 0.1 |
| PIC | 3 | 0.75 | 0.5 | 0.1 |
| COC | 2 | 0.75 | 1 | 0.1 |
| MIX | 2 | 0.75 | 1 | 1 |
| DIA | 1 | 1 | 1 | 1 |
| PHA | 2 | 1 | 1 | 1 |
| **Heterotrophs** | | | | |
| BAC | 4 | 0.1 | 0.1 | 0.1 |
| PRO | 0 | 2 | 1 | 7.5 |
| MES | 0 | 0 | 2 | 10 |
| MAC | 0 | 0 | 0 | 5 |
| JEL | 0 | 0 | 0.5 | 0 |
| **Particulate matter** | | | | |
| Small organic particles | 0.1 | 0.1 | 0.1 | 0.1 |
| Large organic particles | 0.1 | 0.1 | 0.1 | 0.1 |


(2009) dataset, which also included experiments on non-adult and non-*Aurelia* species medusae, unlike the Purcell
et al. (2010) dataset. The relationship between temperature and respiration is heavily skewed by body mass
(Purcell et al., 2010). The data were thus normalised by fitting to a general linear model (GLM) using a least
squares cost function, to reduce the effect of body mass on respiration rates (Ikeda, 1985; Le Quéré et al., 2016).
$GLM = log_{10}RR = a + b\ log_{10}BM + c\ T$           (6)

$$cost\ function = \sum \left( \frac{R_{GLM}^T - R_{obs}^T}{R_{obs}^T} \right)^2 \qquad (7)$$
Where $RR$ is the respiration rate, $BM$ is the body mass, and $T$ and $R^T$ are the observed temperature and associated
respiration rate. The parameters values were then calculated using $R_0 = e^a$, and $Q_{10} = (e^c)^{10}$, where $e$ is the
exponential function. The resulting fit to data is shown in Fig. 3. The parameter values for respiration used in the
model are given in Table 4. Macrozooplankton respiration values are also given in Fig. 3 and Table 4, to provide
a comparison to another zooplankton PFT of the most similar size available.

Table 4. PlankTOM11 parameter values for macrozooplankton and jellyfish, with the associated equation.

| Parameters | JEL | MAC | Equation |
|---|---|---|---|
| Respiration | | | |
| $R_{0°}^{Z_j}$ (d$^{-1}$) | 0.03 | 0.01 | Eq. 1 |
| $d_{Z_j}$ | 1.88 | 2.46 | Eq. 1 |
| Mortality | | | |
| $m_{0°}^{Z_j}$ (d$^{-1}$) | 0.12 | 0.02 | Eq. 1 |
| $c_{Z_j}$ | 1.20 | 3.00 | Eq. 1 |
| $K^{Z_j}$ ($\mu$mol C L$^{-1}$) | 20.0e-6 | 20.0e-6 | Eq. 1 |
| GGE | 0.29 | 0.30 | Eq. 4 |
| Grazing half saturation constant $K_{1/2}^{Z_j}$ ($\mu$mol C L$^{-1}$) | 10.0e-6 | 9.0e-6 | Eq. 5 |


## 2.1.4 Jellyfish PFT Mortality


There is limited data on mortality rates for jellyfish and to use mortality data from the literature on any
zooplankton group some assumptions must be made (Acevedo et al., 2013; Almeda et al., 2013; Malej and Malej,
1992; Moriarty, 2009; Rosa et al., 2013). These assumptions are: that the population is in a steady state where
mortality equals recruitment, reproduction is constant and that mortality is independent of age (Moriarty, 2009).
All models with zooplankton mortality rates follow these assumptions. In reality the mortality of a zooplankton
population is highly variable. Steady states are balanced over a long period (if a population remains viable),
reproduction is restricted to certain times of year and the early stages of life cycles are many times more vulnerable
to mortality. Despite these assumptions, with the limited data on mortality rates, the larger uncertainty lies with
the data rather than the assumptions (Moriarty, 2009). The half saturation constant for mortality ($K_{1/2}^{Zj}$ in Eq. 1) is
set to 20 $\mu$mol C L$^{-1}$ the same as other zooplankton types, due to the lack of PFT specific data. In the small amount
of data available and suitable for use in the model (16 data points from two studies) mortality ranged from 0.006
– 0.026 per day (Acevedo et al., 2013; Malej and Malej, 1992). Applying the exponential fit to these data gave a
mortality rate at 0°C ($m_{0°}^{Zj}$ in Eq. 1) of 0.018 per day. Sensitivity tests were carried out from this mortality rate
due to low confidence in the value.

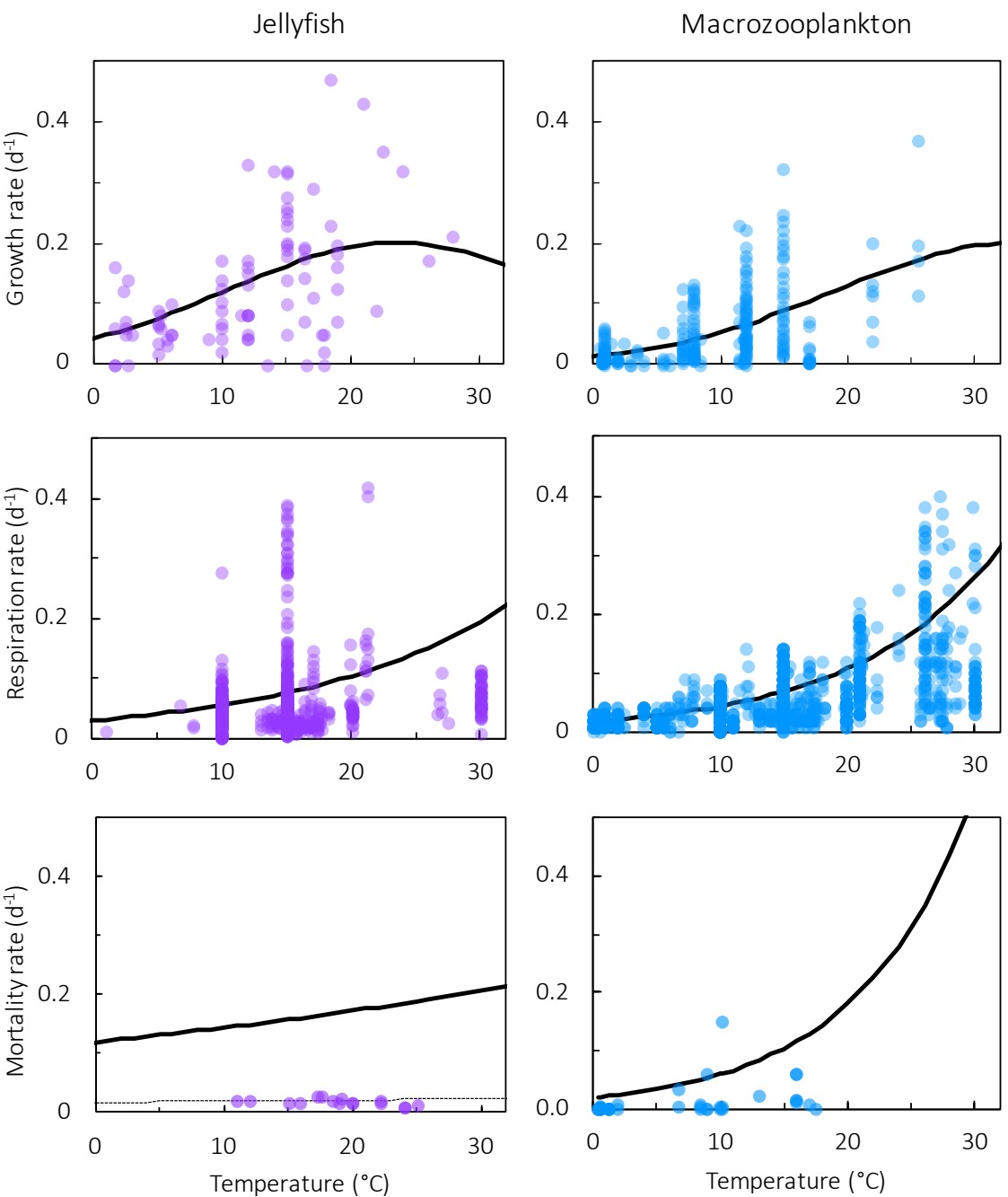

*Figure 3. Maximum growth rates (top), respiration rates (middle) and mortality rates (bottom) for jellyfish (left; purple) and*
*macrozooplankton (right; blue) PFTs as a function of temperature. The fit to data is shown in black, using the parameter*
*values from Table 2 and Table 4. Growth rates are the same as shown in Fig. 2, on a different scale. For jellyfish mortality the*
*thin dashed line is the fit to data and the solid line is the adjusted fit (Table 4).*

Results from a subset of the sensitivity tests are shown in Fig. 4. The model was found to best represent a range of observations when jellyfish mortality was increased to 0.12 per day. The fit to data for mortality ($\mu_0 = 0.018$) and the adjusted mortality ($\mu_0 = 0.12$) is shown in Fig. 3. This value was chosen based on expert judgement of the overall fit across multiple data streams. Whereas it was informed by the quantitative values in Table 6, the final choice required the balance of positive and negative performance that required expert judgement rather than a statistical number. Mortality rate values closer to 0.018 per day allowed jellyfish to dominate macro- and mesozooplankton, greatly reducing their biomass (Fig. 4 and Fig. 5). Low jellyfish mortality also resulted in higher chlorophyll concentrations than observed, especially in the high latitudes (Fig. 4 and Fig. 5; Bar-On et al., 2018; Buitenhuis et al., 2013b). The adjusted mortality rate used for PlankTOM11 may be accounting for several components missing from experimental data including the impact of higher trophic level grazing in the Avecedo et al. (2013) study, which in copepods is 3-4 times higher than other sources of mortality (Hirst and Kiørboe, 2002), the greater vulnerability to mortality experienced during the early stages of the life cycle and mortality due to parasites and viruses, especially during blooms (Pitt et al., 2014).

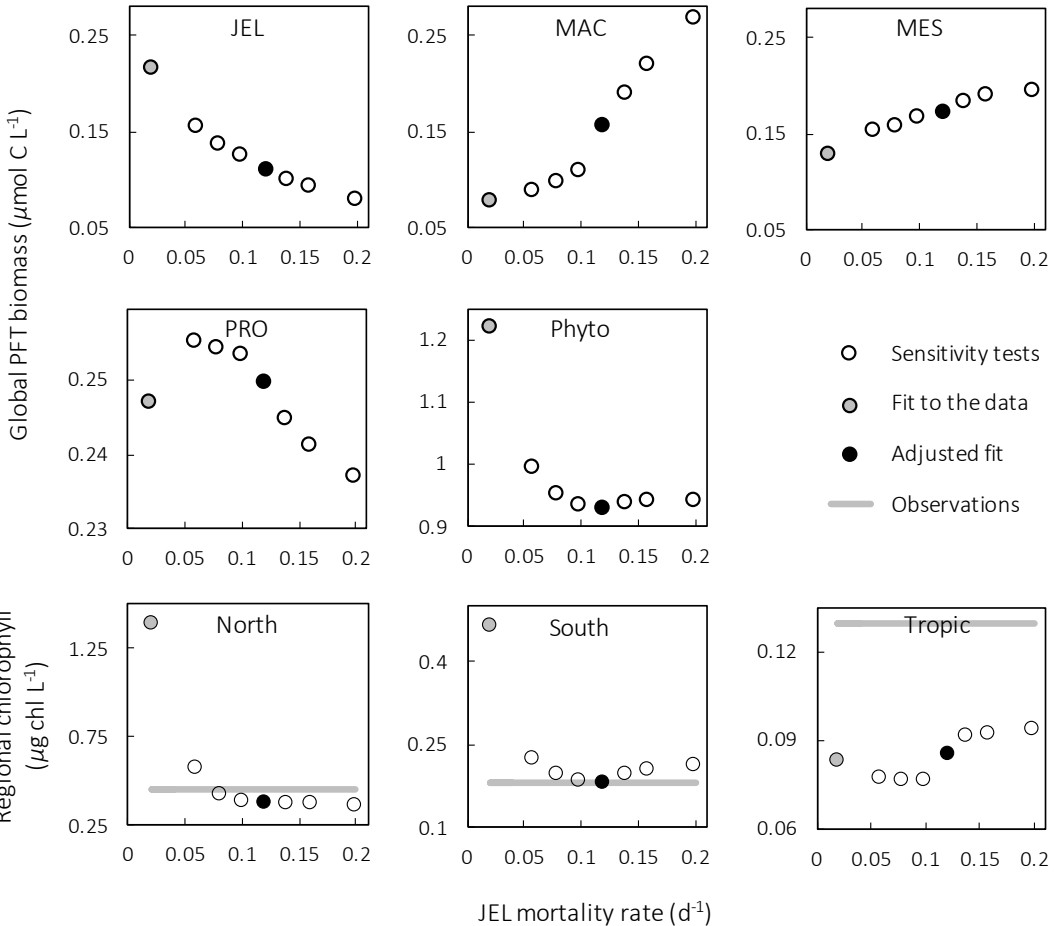

*Figure 4. Results from sensitivity tests on jellyfish mortality rates. The adjusted fit simulation used for PlankTOM11 is shown by the black filled circle and the fit to the data simulation is shown by the grey filled circle; global mean PFT biomass (µmol C L⁻¹) for 0-200m depth (top - middle), regional mean surface chlorophyll concentration (µg chl L⁻¹; bottom). For the regional mean chlorophyll the observations are calculated from SeaWiFS. All data are averaged for 1985-2015, and between 30º and 55º latitude in both hemispheres: 140-240ºE in the north and 140-290ºE in the south (see Fig. 8). Phyto is the sum of all the phytoplankton PFTs.*

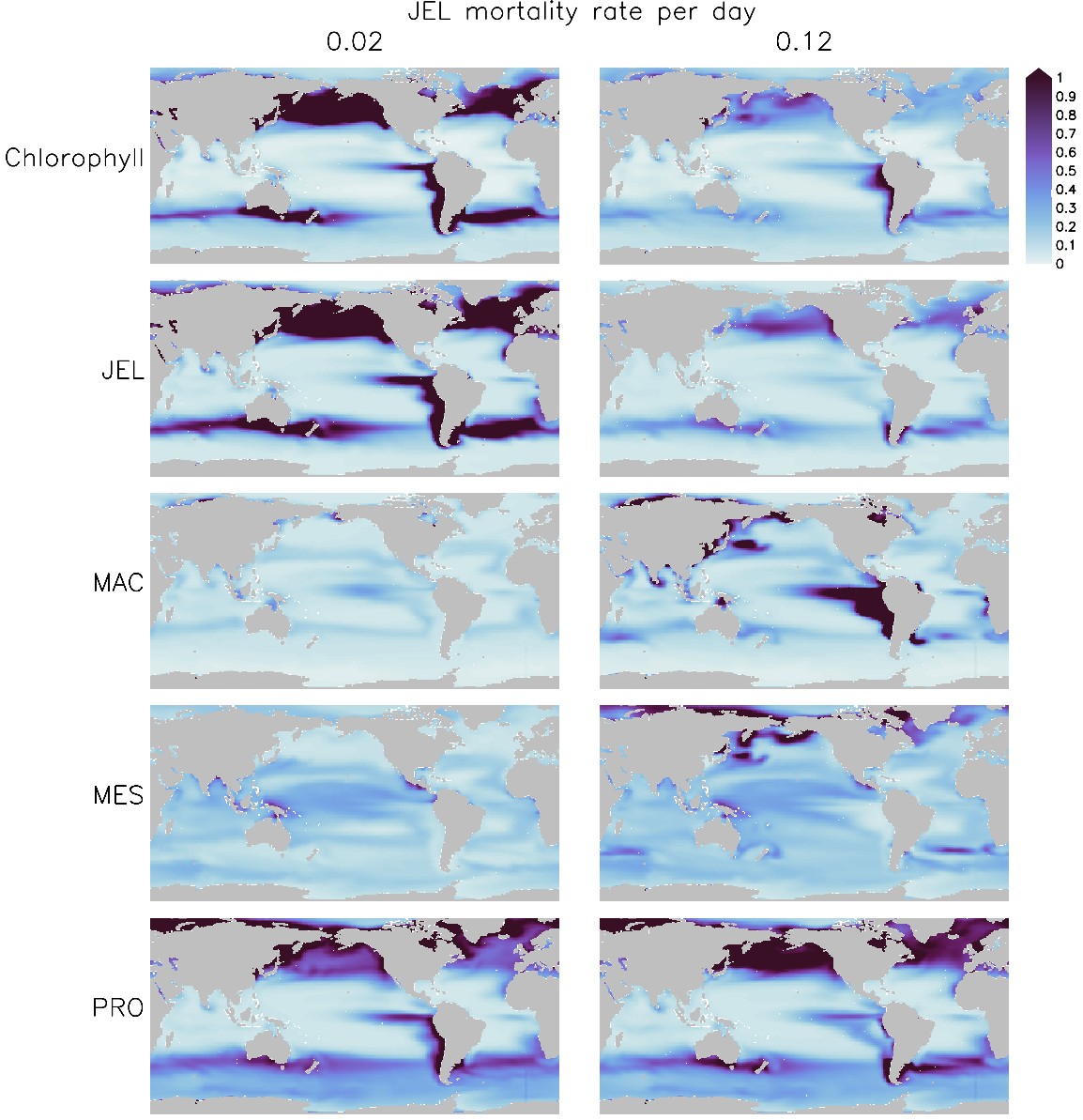

**Figure 5.** *Annual mean surface chlorophyll (µg chl L⁻¹) and zooplankton carbon biomasses (µmol C L⁻¹) of JEL, MAC, MES and PRO for adjustment of JEL mortality for the simulation with 0.02 mortality/day⁻¹ (left) and the adjusted fit simulation with 0.12 mortality/day⁻¹ (right) used in PlankTOM11. Results are shown for the surface box (0-10 meters) and averaged for 1985-2015.*

PlankTOM11 uses a mortality rate for jellyfish that is much higher than the limited observations (Fig. 4 and Fig. 5). Lower jellyfish mortality is likely to be more representative of adult life stages, as jellyfish experience high mortality during juvenile life stages, especially as planula larvae and during settling (Lucas et al., 2012). The limited observations of jellyfish mortality are from mostly adult organisms, which may explain the dominance of jellyfish in the model when parameterised with the observed mortality fit. The higher mortality used for this study may be more representative of an average across all life stages. Experimental jellyfish mortality is also likely to be lower than *in situ* mortality due to factors such as senescence post-spawning and bloom conditions increasing the prevalence of disease and parasites and thus increasing mortality (Mills, 1993; Pitt et al., 2014). Using a higher mortality for this study is therefore deemed reasonable.

## 2.1.5 Organic Carbon Cycling Through the Plankton Ecosystem

In PlankTOM11, the growth of phytoplankton modifies dissolved inorganic carbon into DOC, which then aggregates into $POC_S$ and $POC_L$ (Fig. 1b). $POC_S$ is also generated from protozooplankton egestion and excretion and is consumed through grazing by all zooplankton. $POC_L$ is also generated by aggregation from $POC_S$, egestion and excretion by all zooplankton, and from the mortality of mesozooplankton, macrozooplankton and jellyfish, and is consumed through grazing by all zooplankton. The portion of $POC_S$ and $POC_L$ which is not grazed, sinks through the water column and is counted as export production at 100m (Fig. 1b). The sinking speed of $POC_S$ is 3 $m/d^{-1}$ and the sinking speed of $POC_L$ varies, depending on the concentration of ballast and the resulting particle density. Proto-, meso- and macrozooplankton excretion is largely in the form of particulate and solid faecal pellets, while this makes up very little of jellyfish excretion. Jellyfish instead produce and slough off mucus as part of their feeding mechanism (Pitt et al., 2009), which is represented in the model in the same way as the faecal pellet excretion, as a fraction of unassimilated grazing contributing to $POC_L$.

## 2.1.6 Additional Tuning

Following the change to the growth rate formulation (from Eq. 2 to Eq. 3), all PFT growth rates are lower compared to the published version of PlankTOM10 (Le Quéré et al., 2016), but the change is largest for *Phaeocystis*, diatoms, bacteria and protozooplankton (Fig. 2). Further tuning is carried out to rebalance the total biomass among phytoplankton PFTs following the change in formulation. The tuning included increasing the grazing ratio preference of mesozooplankton for *Phaeocystis* and the grazing ratio preference of protozooplankton for picophytoplankton within the limits of observations. Tuning also included increasing the half saturation constant of the phytoplankton *Phaeocystis*, picophytoplankton and diatoms for iron. The tuning resulted in a reduction of *Phaeocystis* biomass and an increase in diatom biomass, without disrupting the rest of the ecosystem. Diatom respiration was also increased to reduce their biomass towards observations. Finally, bacterial biomass was increased closer to observations by reducing the half saturation constant of bacteria for dissolved organic carbon and reducing the maximum bacteria uptake rate. See Appendix Table A4 for the parameter values before and after tuning.

As shown in Eq. 1, there is a component in the mortality of zooplankton to represent predation by organisms not included in the model. The jellyfish PFT is a significant grazer of macrozooplankton and mesozooplankton (Table 3), to account for this additional grazing the mortality term for macrozooplankton and the respiration term for mesozooplankton were reduced compared to model versions where no jellyfish are present (Table 5). Respiration is reduced in place of mortality for mesozooplankton as their mortality term had already been reduced to zero to account for predation by macrozooplankton (Le Quéré et al., 2016). The jellyfish PFT is also a significant grazer of protozooplankton, however, following the adjustment of protozooplankton grazing on picophytoplankton to

account for changes to the growth rate formulation and the low sensitivity of protozooplankton to jellyfish mortality (Fig. 4) additional changes to protozooplankton parameters were found to be unnecessary.

**Table 5.** Changes to non-jellyfish PFT parameters across the PlankTOM simulations. PlankTOM10$^{LQ16}$ is the latest published version of PlankTOM with 10 PFTs (Le Quéré et al., 2016), while PlankTOM10 is the simulation from this study.

| Parameters | PlankTOM10$^{LQ16}$ | PlankTOM10 | PlankTOM10.5 | PlankTOM11 |
|---|---|---|---|---|
| MAC mortality | 0.020 | 0.012 | 0.005 | 0.005 |
| MES respiration | 0.014 | 0.014 | 0.001 | 0.001 |

## 2.1.7 Model Simulations

The PlankTOM11 simulations are run from 1920 to 2015, forced by meteorological data including daily wind stress, cloud cover, precipitation and freshwater riverine input from NCEP/NCAR reanalysed fields (Kalnay et al., 1996). The simulations start with a 28-year spin for 1920-1948 where the meteorological conditions for year 1980 are used, looping over a single year. Year 1980 is used as a typical average year, as it has no strong El Nino/La Nina, as in Le Quéré et al. (2010). Furthermore, because of the greater availability of weather data (including by satellite) in 1980 compared to 1948, the dynamical fields are generally more representative of small-scale structures than the earlier years. There is a small shock to the system at the start of meteorological forcing, but this stabilises within a few years and decades before the model output is used for analysis. Tests of different spin-up years were carried out in Le Quéré et al. (2010), including both 1948 and 1980, with little impact on trends generally. The spin up is followed by interannually varying forcing for actual years from 1948-2015. All analysis is carried out on the average of the last 31-year period of 1985-2015. PlankTOM11 is initialised with observations of dissolved inorganic carbon (DIC) and alkalinity (Key et al., 2004) after removing the anthropogenic component for DIC (Le Quéré et al., 2010), $NO_3$, $PO_4$, $SiO_3$, $O_2$, temperature and salinity from the World Ocean Atlas (Antonov et al., 2010).

Two further model simulations were carried out in order to better understand the effect of adding the jellyfish PFT. The first simulation sets the jellyfish growth rate to 0, so that it replicates the model set up with 10 PFTs in Le Quéré et al. (2016), here called PlankTOM10$^{LQ16}$, but it includes the updated growth formulation (Sect. 2.1.1) and additional tuning (Sect. 2.1.5). The simulation is labelled 'PlankTOM10' in the figures. This simulation is otherwise identical to PlankTOM11 except for the mortality term for macrozooplankton and the respiration term for mesozooplankton, which were initially returned to PlankTOM10$^{LQ16}$ values, to account for the lack of predation by jellyfish. Macrozooplankton mortality was then tuned down from the PlankTOM10$^{LQ16}$ value, from 0.02 to 0.012, to account for the change to the growth calculation (Table 5). The second additional simulation is carried out to test the addition of an 11$^{th}$ PFT in comparison to the addition of jellyfish as the 11$^{th}$ PFT. This is done by parameterising the jellyfish PFT identically to the macrozooplankton PFT in PlankTOM11, so that there are 11 PFTs active, with two identical macrozooplankton. This simulation is called PlankTOM10.5. The two

macrozooplankton in PlankTOM10.5 have mutual predation, where they prey on each other, while the macrozooplankton in PlankTOM10 have no preference for themselves. Subsequently, macrozooplankton mortality in PlankTOM10.5 is kept the same as PlankTOM11 (Table 5) to account for the mutual predation. Otherwise, these simulations were identical to PlankTOM11.

## 2.2 JELLYFISH BIOMASS OBSERVATIONS

MARine Ecosystem biomass DATa (MAREDAT) is a database of global ocean plankton abundance and biomass, harmonised to common units and is open source available online (Buitenhuis et al., 2013b). The MAREDAT database is designed to be used for the validation of global ocean biogeochemical models. MAREDAT contains global quantitative observations of jellyfish abundance and biomass as part of the generic macrozooplankton group (Moriarty et al., 2013). The jellyfish sub-set of data has not been analysed independently yet.

For this study, all MAREDAT records under the group Cnidaria medusae ('true' jellyfish) were extracted from the macrozooplankton group (Moriarty et al., 2013) and examined. The taxonomic level within the database varies from phylum down to species. The data covers the period from August 1930 to August 2008 and contains abundance (individuals/m$^3$, n=107,156) and carbon biomass ($\mu$g carbon L$^{-1}$, n=3,406). The carbon biomass data are used over the abundance data despite the fewer data available, as they can be directly compared to PlankTOM11 results. Carbon biomass is calculated from wet weight/dry weight conversion factors for species where data records are sufficient (Moriarty et al., 2013). The data were collected at depth ranging from 0 to 2442m. The majority of the data (97%) were collected in the top 200m with an average depth of 44m ($\pm$ 32m). Data from the top 200m are included in the analysis. The original un-gridded biomass data were binned into 1ºx1º degree boxes at monthly resolution, as in Moriarty et al. (2013), reducing the number of gridded biomass data points to 849.

In MAREDAT, jellyfish biomass data are only present in the Northern Hemisphere, which is likely to skew the data. Another caveat to the data is that a substantially smaller frequency of zeros is reported for biomass than for abundance. Under-reporting of zero values will increase the average, regardless of the averaging method used. Biomass observations from other global studies (Bar-On et al., 2018; Lucas et al., 2014; Luo et al., 2020) are used conjunctly with the global jellyfish biomass calculated here because of the poor spatial coverage.

To compare to the other PFTs within the MAREDAT database, global jellyfish biomass was calculated according to the methods in Buitenhuis et al. (2013b). Buitenhuis et al. (2013b) calculate a biomass range, using the median as the minimum and the arithmetic mean (AM) as the maximum. The jellyfish zooplankton biomass range in MAREDAT was calculated as 0.46 – 3.11 PgC, with the median jellyfish biomass almost as high as the microzooplankton and higher than meso- and macrozooplankton (Buitenhuis et al., 2013b). The jellyfish biomass range calculated here is used to validate the new jellyfish component in the PlankTOM11 model.

# 3 RESULTS

## 3.1 JELLYFISH BIOMASS

The global jellyfish biomass estimated by various studies gives a range of results: 0.1 PgC (Bar-On et al., 2018), 0.32 ± 0.49 PgC (Lucas et al., 2014), 0.29 ± 0.56 PgC (Luo et al., 2020, updated from Lucas et al.) and 0.46 – 3.11 PgC calculated in this study (Sect. 2.2). Jellyfish biomass in PlankTOM11 is within the range but towards the lower end of observations at 0.13 PgC, with jellyfish accounting for 16% of the total zooplankton biomass (Table 6). When the modelled biomass was tuned to match the higher observed biomass by adjusting the mortality rate, jellyfish dominate the entire ecosystem significantly reducing levels of the other zooplankton and increasing chlorophyll above observations for the Northern and Southern Hemispheres (Fig. 4 and Fig. 5).

PlankTOM11 generally replicates the patterns of jellyfish biomass with observations. High biomass occurs at around 50-60°N across the oceans, with the highest biomass in the North Pacific. PlankTOM11 also replicates low biomass in the Indian Ocean, and the eastern half of the tropical Pacific shows higher biomass than other open ocean areas in agreement with patterns in observations (Fig. 6; Lucas et al., 2014; Luo et al., 2020). However, PlankTOM11 underestimates the high jellyfish biomass in the tropical Pacific (Fig. 6). Most of the data informing the jellyfish parameters is from temperate species, so the model will better represent higher latitudes than lower latitudes. This is likely responsible for some of the underestimation of biomass in this region. The competition of jellyfish with macrozooplankton also plays a role (see Sect. 3.3 for further discussion). The lack of biomass observations around 40°S makes it difficult to determine if the peak in jellyfish biomass in PlankTOM11 at this latitude is representative of reality. The maximum biomass in the southern hemisphere is mostly around coastal areas i.e. South America and southern Australia (Fig. 6). This is expected from reports and papers on jellyfish in these areas (Condon et al., 2013; Purcell et al., 2007 and references therein). A prevalence of jellyfish in coastal areas is apparent (Fig. 6), in line with observations (Lucas et al., 2014; Luo et al., 2020), even without any specific coastal advantages for jellyfish in the model (see macrozooplankton in Le Quéré et al., 2016). However, PlankTOM11 underestimates the range of observations in the top 200m (Fig. 6). PlankTOM11 overestimates the minimum values and underestimates the maximum values. However, part of this discrepancy may be due to under-sampling in the observations. A key caveat in jellyfish data is that the data is not uniformly distributed spatially or temporally and not proportionally distributed between various biomes of the ocean, with collection efforts skewed to coastal regions and the Northern Hemisphere (MAREDAT; Lilley et al., 2011; Lucas et al., 2014; Luo et al., 2020). This sampling bias and sampling methods also tend to favour larger, less delicate species, which are often scyphomedusae with a meroplanktonic life cycle.

Jellyfish are characterised by their bloom and bust dynamic, resulting in patchy and ephemeral biomass. The mean:max biomass ratio of observations (MAREDAT) was compared to the same ratio for PlankTOM11 to assess the replication of this characteristic. The observations give a wide range of ratios depending on the type of mean used. The PlankTOM11 ratio falls within this range, but towards the lower end (Table 7). PlankTOM11 replicates some of the patchy and ephemeral biomass of jellyfish.

Jellyfish biomass in MAREDAT has poor global spatial coverage. The region around the coast of Alaska has the highest density of observations and is used here to evaluate the mean, range and seasonality of the carbon biomass of jellyfish as represented in PlankTOM11. The gridded jellyfish observations from Luo et al., (2020; see Fig. 6)

**Table 6.** Global mean values for rates and biomass from observations and the PlankTOM11 and PlankTOM10 models averaged over 1985–2015. In parenthesis is the percentage share of the plankton type of the total phytoplankton or zooplankton biomass. The percentage share of mixed-phytoplankton is not included, as there are no mixed-phytoplankton observations, therefore, the phytoplankton percentages are of total phytoplankton minus mixed-phytoplankton. References for observations are given in Appendix Table A5.

| | PlankTOM11 | PlankTOM10 | Observations |
|---|---|---|---|
| **Rates** | | | |
| Primary production (PgC $y^{-1}$) | 41.6 | 43.4 | 51-65 |
| Export production at 100m (PgC $y^{-1}$) | 7.1 | 7.0 | 5-13 |
| $CaCO_3$ export at 100m (PgC $y^{-1}$) | 1.3 | 1.2 | 0.6-1.1 |
| $N_2$ fixation (TgN $y^{-1}$) | 97.2 | 95.9 | 60-200 |
| **Phytoplankton biomass 0-200m (PgC)** | | | |
| $N_2$-fixers | 0.065 (8%) | 0.075 (10%) | 0.008-0.12 (2-8%) |
| Picophytoplankton | 0.141 (17%) | 0.153 (20%) | 0.28-0.52 (35-68%) |
| Coccolithophores | 0.248 (30%) | 0.212 (27%) | 0.001-0.032 (0.2-2%) |
| Mixed-phytoplankton | 0.263 | 0.268 | - |
| Phaeocystis | 0.177 (22%) | 0.170 (22%) | 0.11-0.69 (27-46%) |
| Diatoms | 0.183 (22%) | 0.167 (21%) | 0.013-0.75 (3-50%) |
| Total phytoplankton biomass | 1.077 | 1.046 | 0.412 – 2.112 |
| **Heterotrophs biomass 0-200m (PgC)** | | | |
| Bacteria | 0.041 | 0.046 | 0.25-0.26 |
| Protozooplankton | 0.295 (36%) | 0.330 (32.7%) | 0.10-0.37 (27-31%) |
| Mesozooplankton | 0.193 (23%) | 0.218 (21.6%) | 0.21-0.34 (25-66%) |
| Macrozooplankton | 0.205 (25%) | 0.460 (45.6%) | 0.01-0.64 (3-47%) |
| Jellyfish zooplankton | 0.129 (16%) | - | 0.10-3.11 |
| Total zooplankton biomass | 0.823 | 1.008 | 0.42 – 4.46 |

are available as a mean over time and depth, so cannot be used to evaluate range or seasonality. Spatially, the

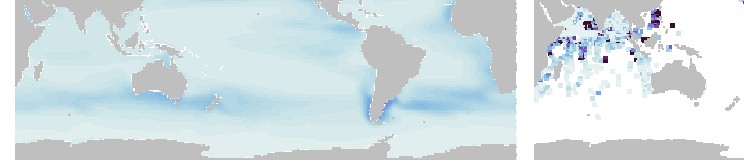

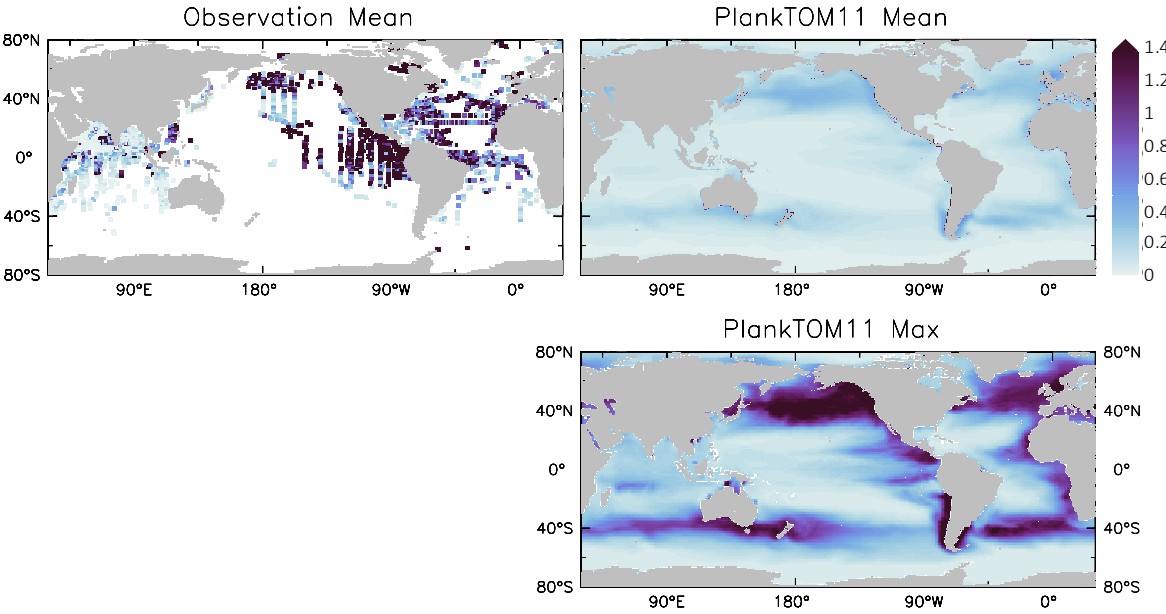

*Figure 6. Jellyfish carbon biomass (μmol C L⁻¹) in PlankTOM11 and in observations from the Jellyfish Database Initiative (Luo*
*et al., 2020). PlankTOM11 results (left) are the mean and maximum biomass from monthly climatologies. Observations (right)*
*are the mean biomass, areas with no observations are in white. Observations are on a 1x1° grid and are plotted using a*
*three-cell averaging filler for visual clarity. All data is for 0-200m. The gridded observation data is only available as a mean*
*over time and depth (Luo et al., 2020). Due to the patchy nature of the observations in depth and time, the mean may be*
*skewed high or low, while the model is sampled across the full time and depth.*
observations peak around the north coast of Alaska while PlankTOM11 peaks around the south coast (Fig. 7).
This difference is likely due to the lack of small-scale physical processes in the model due to the relatively coarse
model resolution. PlankTOM11 reproduces the observed mean jellyfish biomass around the coast of Alaska (0.16
compared to 0.13 $\mu$mol C L⁻¹), but it underestimates the maximum and spread of the observations (Table 8). The
spatial patchiness is somewhat replicated in PlankTOM11, although with a smaller variation (Fig. 7).
PlankTOM11 replicates the mean seasonal shape and biomass of jellyfish with a small peak over the summer
followed by a large peak in September in the observations and in October in PlankTOM11 (Fig. 7). Overall,
PlankTOM11 replicates the mean but underestimates the maximum biomass and temporal patchiness of the
observations (Fig. 7 and Table 8).

**Table 7.** Jellyfish biomass globally from observations (MAREDAT) and PlankTOM11. Three types of mean are given for the observations; Med is the median, AM is the arithmetic mean and GM is the geometric mean. The ratios are all scaled to mean = 1. All units are $\mu$g C L⁻¹.

|              |     | Mean | Max   | Ratio   |
|--------------|-----|------|-------|---------|
| Observations | AM  | 3.61 | 156.0 | 1 : 43  |
|              | GM  | 0.95 | 156.0 | 1 : 165 |
|              | Med | 0.29 | 156.0 | 1 : 538 |
| PlankTOM11   | AM  | 1.18 | 98.9  | 1 : 84  |

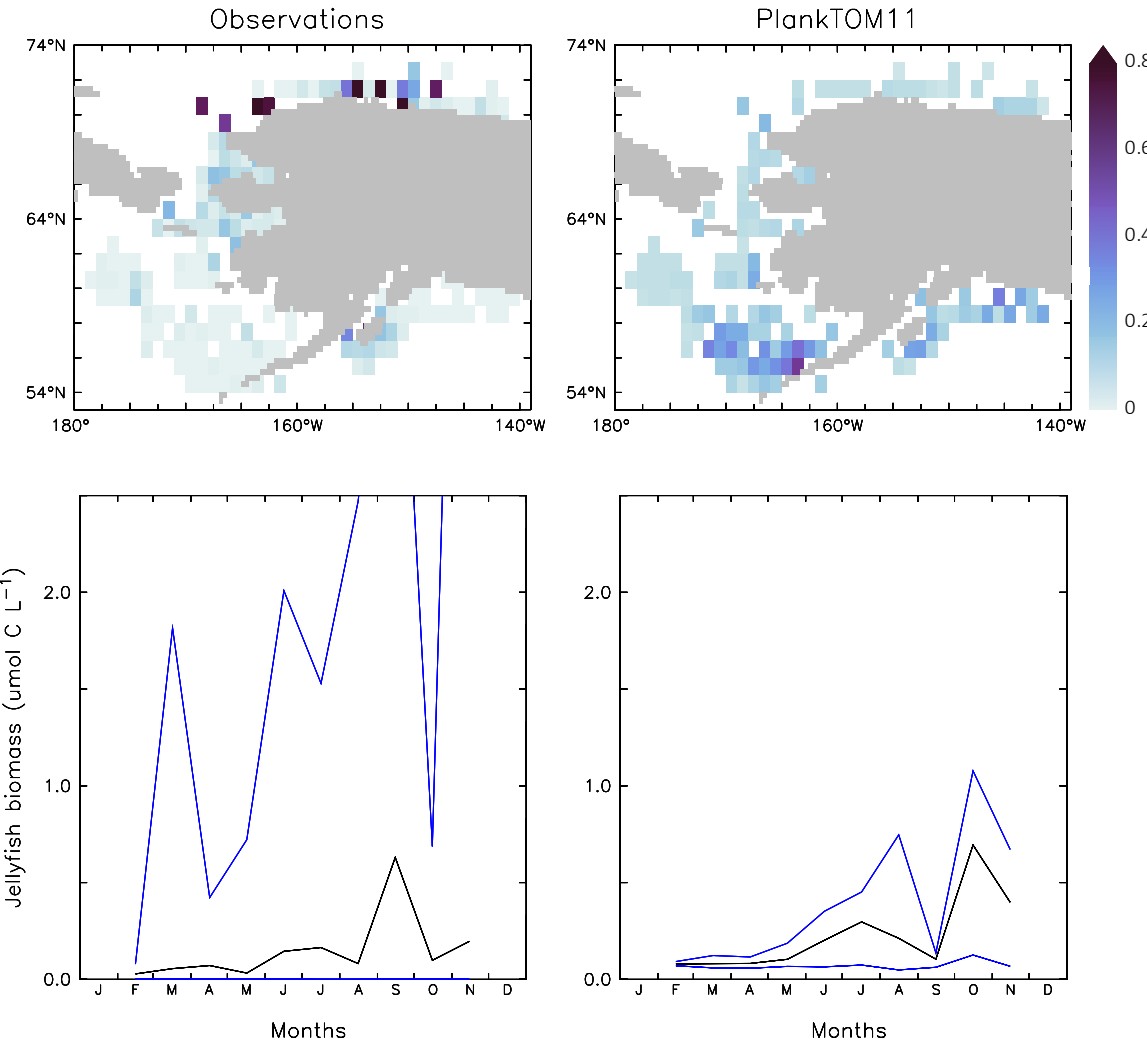


*Figure 7. Carbon biomass of jellyfish (μmol C L⁻¹) from MAREDAT observations (left) and PlankTOM11 (right) for the coast of*
*Alaska (the region with the highest density of observations). The top panels show the mean jellyfish biomass and the bottom*
*panels show the seasonal jellyfish biomass, with the monthly mean in black and the monthly minimum and maximum in blue.*
*Observations and PlankTOM11 results are for 0-150m, as the depth range where >90% of the observations occur. No*
*observations were available for January or December.*

436

## 3.2 ECOSYSTEM PROPERTIES OF PLANKTOM11

438

PlankTOM11 reproduces the main characteristics of surface chlorophyll observations, with high chlorophyll
concentration in the high latitudes, low concentration in the subtropics and elevated concentrations around the
equator (Fig. 8). PlankTOM11 also reproduces higher chlorophyll concentrations in the Northern Pacific than the
Southern (Fig. 9), and higher concentrations in the southern Atlantic than the southern Pacific Ocean (Fig. 8).
Overall the model underestimates chlorophyll concentrations, as is standard with models of this type (Le Quéré
et al., 2016) particularly in the central and northern Atlantic. PlankTOM11 also captures the seasonality of
chlorophyll, with concentrations increasing in summer compared to the winter for each hemisphere (Fig. 8).

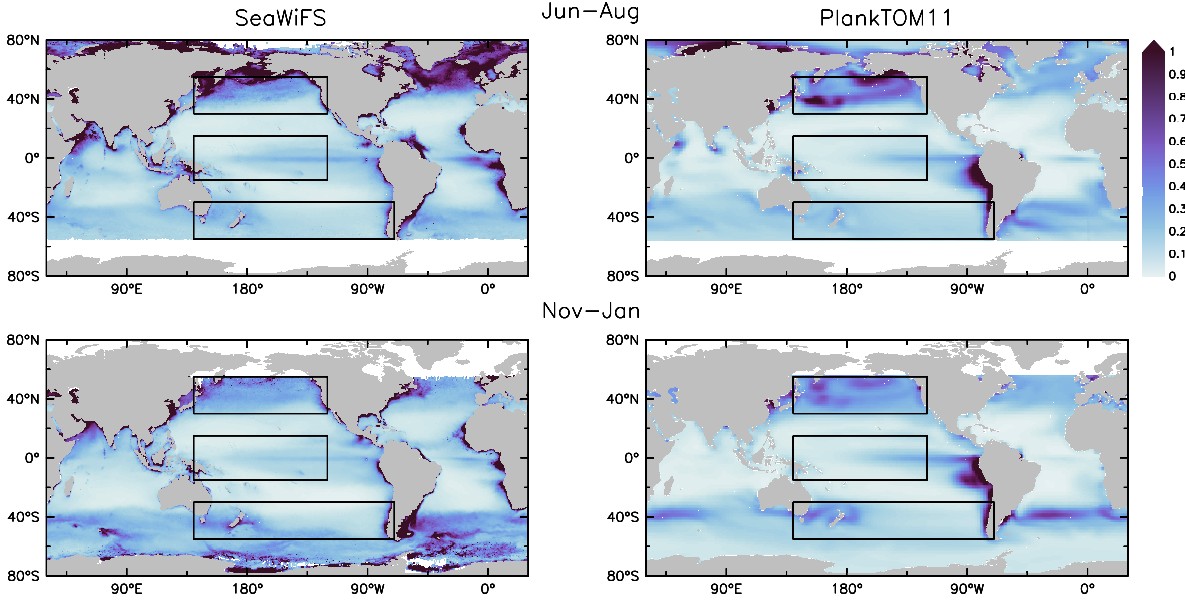


**Figure 8.** *Surface chlorophyll (µg chl L⁻¹) averaged for June to August (top) and November to January (bottom). Panels show observations from SeaWiFS (left) satellite and results from PlankTOM11 (right). Observations and model are averaged for 1997-2006. The black boxes show the Pacific north, tropic and south regions used in Fig. 4 and Fig. 9.*


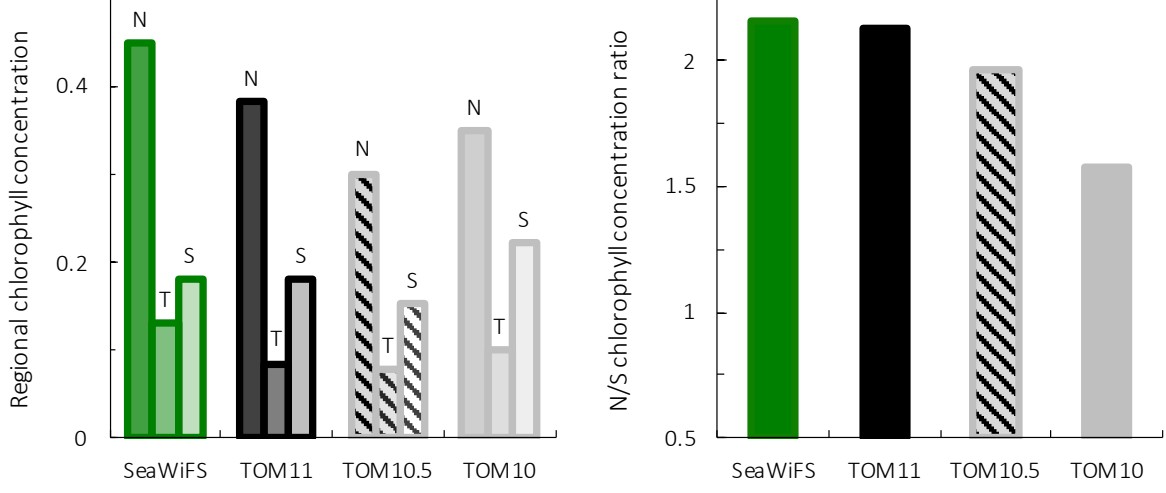


**Figure 9.** *Surface chlorophyll for observations from SeaWiFS satellite, PlankTOM11, PlankTOM10.5 and PlankTOM10. Regional chlorophyll concentration in µg chl L⁻¹ (right) for the north (N), tropic (T) and south (S) Pacific Ocean regions shown in Fig. 8 and the N/S chlorophyll concentration ratio (left). Observations and model are averaged for 1997-2006.*


To assess the effect of adding jellyfish to PlankTOM, two additional simulations were conducted: PlankTOM10 where jellyfish growth is set to zero and PlankTOM10.5 where all jellyfish parameters are set equal to macrozooplankton parameters (Sect. 2.1.6). The two simulations show similar spatial patterns of surface chlorophyll to PlankTOM11, but different concentration levels. PlankTOM11 closely replicates the chlorophyll ratio between the north and south Pacific with a ratio of 2.12, compared to the observed ratio of 2.16 (Fig. 9).

PlankTOM10 and PlankTOM10.5 underestimate the observed ratio with ratios of 1.57 and 1.96 respectively (Fig.
9). Adding an 11th PFT improves the chlorophyll ratio, however, the regional chlorophyll concentrations for
PlankTOM10.5 are a poorer match to the observations than PlankTOM11, especially in the north (Fig. 9).
PlankTOM10 overestimates the observed chlorophyll concentration in the south (0.22 and 0.18 respectively; Fig.
9). All three simulations underestimate chlorophyll concentration in the tropics compared to observations (Fig.
9). The north/south chlorophyll ratio metric was developed by Le Quéré et al. (2016) as a simple method to
quantify model performance for emergent properties, focussing on the Pacific Ocean as the area where this ratio
is most pronounced in the observations. These simulations further support the suggestion by Le Quéré et al. (2016)
that the observed distribution of chlorophyll in the north and south is a consequence of trophic balances between
the PFTs and improves with increasing plankton complexity.
PlankTOM11 underestimates primary production by 10 PgC $y^{-1}$, which is similar to the underestimation in
PlankTOM10$^{LQ16}$ of 9 PgC $y^{-1}$. As suggested by Le Quéré et al. (2016) this may be due to the model only
representing highly active bacteria, which is unchanged between the model versions, while observed biomass is
also from low activity bacteria and ghost cells. Export production and $N_2$ fixation are within the observational
range, and $CaCO_3$ export is slightly overestimated (Table 6).
In PlankTOM11 each PFT shows unique spatial distribution in carbon biomass (Fig. 5). The total biomass of
phytoplankton is within the range of observations, but the partitioning of this biomass between phytoplankton
types differs from observations (Table 6). PlankTOM11 is dominated by mixed-phytoplankton and
coccolithophores, together making up 47% of the total phytoplankton biomass. Diatoms and *Phaeocystis* are the
next most abundant and fall within the observed range, followed by picophytoplankton with around half the
observed biomass (Table 6). The observations are dominated by picophytoplankton, followed by *Phaeocystis* and
Diatoms (Table 6). The modelled mixed-phytoplankton is likely taking up the ecosystem niche of
picophytoplankton. Coccolithophores are overestimated by a factor of 10 and may also be filling the ecosystem
niche of picophytoplankton in the model (Table 6). The phytoplankton community composition changed from
PlankTOM10$^{LQ16}$ to PlankTOM11, with some phytoplankton types moving closer to observations and some
moving further away. For example, for $N_2$-fixers PlankTOM11 is in line with the upper end of observations at
8%, while PlankTOM10 and PlankTOM10$^{LQ16}$ overestimate $N_2$-fixers (10% and 11% respectively). For
picophytoplankton, PlankTOM10$^{LQ16}$ is within the range of observations at 38%, while PlankTOM11 and
PlankTOM10 underestimate the community share of picophytoplankton (17% and 20% respectively). For
*Phaeocystis*, all three simulations underestimate the community share, but PlankTOM11 and PlankTOM10 (both
22%) are closer to the lower end of observations (27%) than PlankTOM10$^{LQ16}$ (15%; Table 6; Le Quéré et al,
2016). Overall, the difference between PlankTOM10$^{LQ16}$ and PlankTOM11 is greater than the difference between
PlankTOM10 and PlankTOM11, suggesting that the change to growth of PFT's had a larger effect on
phytoplankton community composition than the addition of jellyfish. This is expected, as the growth change
directly effects each PFT and model results are sensitive to PFT growth rates (Buitenhuis et al., 2006, 2010).
Jellyfish affect phytoplankton community composition, but the effect is small.

## 3.3 ROLE OF JELLYFISH IN THE PLANKTON ECOSYSTEM

Macrozooplankton exhibit the largest change in biomass between the three simulations, followed by mesozooplankton (Fig. 10). This is despite the higher preference of jellyfish grazing on mesozooplankton (ratio of 10) than on macrozooplankton (ratio of 5; Table 3). The central competition for resources between jellyfish and macrozooplankton is that they both preferentially graze on mesozooplankton, then on protozooplankton, although macrozooplankton have a lower preference ratio for zooplankton than jellyfish, as more of their diet is made up by phytoplankton (Table 3). In simple terms this means that for two equally sized populations of jellyfish and macrozooplankton, jellyfish would consume more meso- and protozooplankton than would be consumed by macrozooplankton. However, predator biomass, prey biomass and the temperature dependence of grazing interact to affect the rate of consumption (Eq. 5). The greatest difference in PFT biomass, especially macrozooplankton biomass, between simulations occurs in latitudes higher than 30º where jellyfish biomass is highest (Fig. 10). In the tropics, jellyfish have a low impact on the ecosystem due to their low biomass in this region (Fig. 6 and Fig. 10).

The seasonality of the PFTs in each simulation is shown in Fig. 11 for 30-70º north and south, as the regions with the greatest differences between simulations (Fig. 10). In PlankTOM10 macrozooplankton represent the highest trophic level. The addition of another PFT at the same or at a higher trophic level (PlankTOM10.5 and PlankTOM11 respectively) reduces the biomass of the macrozooplankton, through a combination of competition and low-level predation (Fig. 10 and Fig. 11). For PlankTOM10.5 results, macrozooplankton is summed with the 11th PFT (identical to macrozooplankton in this simulation). The addition of this 11th PFT at the same trophic level reduces the biomass of the macrozooplankton (Fig. 10 and Fig. 11), despite the macrozooplankton mortality being reduced from PlankTOM10 to PlankTOM10.5 (Table 5) which would be expected to increase macrozooplankton biomass. However, the low level of mutual predation between the two macrozooplankton PFTs slightly reduces their overall biomass. This reduction in biomass mostly occurs during the autumn macrozooplankton bloom, where the peak is reduced from PlankTOM10 to PlankTOM10.5, while the winter – spring biomass is similar across the two simulations (Fig. 11). The drop in mesozooplankton respiration from PlankTOM10 to PlankTOM10.5 (Table 5) lowers the rate of respiration, especially at lower temperatures. This likely accounts for the increase in PlankTOM10.5 mesozooplankton biomass at higher latitudes (Fig. 10). The addition of jellyfish changes the zooplankton with the highest biomass from macrozooplankton to protozooplankton and reduces the biomass of mesozooplankton, in both the north and south (Fig. 11). However, the impact on the biomass of mesozooplankton and protozooplankton is small, despite mesozooplankton being the preferential prey of jellyfish, followed by protozooplankton. The small impact of jellyfish on mesozooplankton and protozooplankton biomass may be due to trophic cascade effects where jellyfish reduce the biomass of macrozooplankton, which reduces the predation pressure of macrozooplankton on meso- and protozooplankton, whilst jellyfish simultaneously provide an additional predation pressure on meso- and protozooplankton. The decrease in predation by macrozooplankton may be compensated for by the increase in predation by jellyfish, resulting in only a small change to the overall biomass of mesozooplankton and protozooplankton.

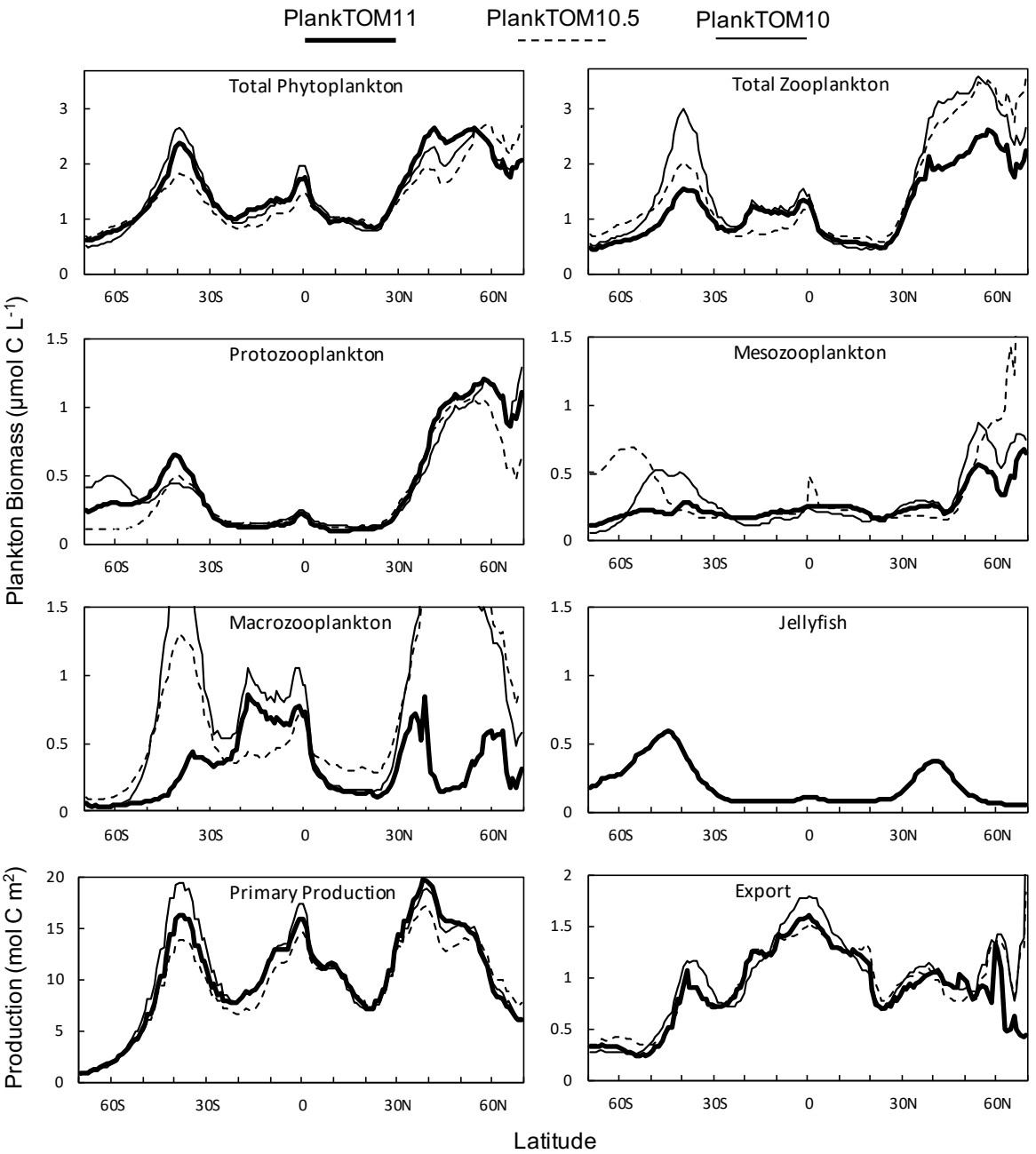

**Figure 10.** *Zonal mean distribution for the PlankTOM11, PlankTOM10.5 and PlankTOM10 simulations. All plankton biomass data are for the surface box (0-10m). For PlankTOM10.5 the MAC PFT has been summed with the 11th PFT that duplicates MAC. The bottom panels are the zonal mean distribution of primary production, integrated over the top 100m, and export production at 100m. All data are averaged for 1985-2015.*

In PlankTOM11 there is a clear distinction between the biomass in the north and south, with higher biomass for each PFT in the north compared to the south (Fig. 10 and Fig. 11). Plankton types have higher concentrations in the respective hemisphere's summer, and a double peak in phytoplankton in the north (Fig. 11). PlankTOM10 also has a higher biomass of each PFT in the north compared to the south, but the difference is smaller than that in PlankTOM11 (Fig. 10 and Fig. 11). The key difference between the two models is the biomass of macrozooplankton. In PlankTOM10 macrozooplankton are the dominant zooplankton, especially in late summer and autumn where their biomass matches and even exceeds the biomass of phytoplankton in the region (Fig. 11). In PlankTOM11 neither macrozooplankton, nor any other zooplankton, come close to matching the biomass of

phytoplankton. The largest direct influence of jellyfish in these regions is its role in controlling macrozooplankton biomass, through competition for prey resources, particularly mesozooplankton and protozooplankton, and through the predation of jellyfish on macrozooplankton.

In PlankTOM11 in the north, phytoplankton display a double peak in seasonal biomass, with a smaller peak in April of 2.9 $\mu$mol C L$^{-1}$, followed by a larger peak in July of 3.2 $\mu$mol C L$^{-1}$ (Fig. 11). The addition of jellyfish amplifies these peaks from PlankTOM10 and PlankTOM10.5 (Fig. 11) and from PlankTOM10 (Le Quéré et al., 2016). Observations (MAREDAT) show two peaks in phytoplankton biomass although the peaks are offset in timing from all three PlankTOM simulations. The amplitude of the full seasonal cycle in observations is 0.78 – 2.67 µmol C/L (median – mean) with all three PlankTOM simulations falling well within this range (Table A6). Removing the winter months, where there is less variability, gives a non-winter observational amplitude of 0.7 – 2.12 µmol C/L. PlankTOM11 is the highest, with a non-winter amplitude of 0.97 µmol C/L, with the other two simulations lower at 0.8 µmol C/L (PLankTOM10.5) and 0.81 µmol C/L (PlankTOM10; Table A6). PlankTOM10$^{LQ16}$ has a lower seasonal amplitude than PlankTOM11, although a slighter higher non-winter amplitude by 0.05 µmol C/L (Table A6). The changes to phytoplankton seasonal biomass are not evenly distributed across the PFT's, with coccolithophores and Phaeocystis exhibiting the largest changes (Fig. A1).

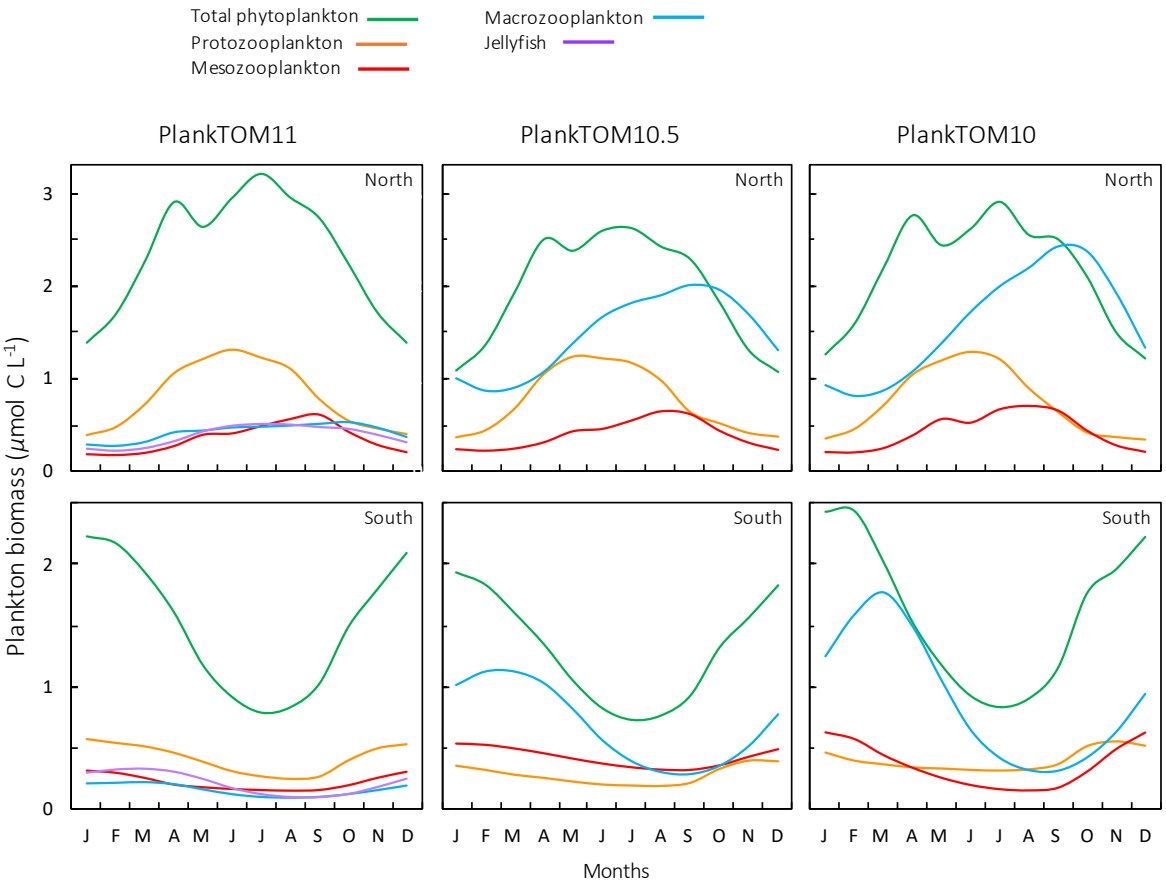

**Figure 11.** *Seasonal surface carbon biomass (µmol C L$^{-1}$) of total phytoplankton PFTs, protozooplankton, mesozooplankton, macrozooplankton and jellyfish. For PlankTOM10.5 the MAC PFT has been summed with the 11$^{th}$ PFT that duplicates MAC. Panels shown PFT biomass for PlankTOM11 (left), PlankTOM10.5 (middle) and PlankTOM10 (right), for two regions; the north 30ºN - 70ºN (top) and the south 30ºS - 70ºS (bottom) across all longitudes. All data are averaged for 1985-2015.*

Primary production follows a similar pattern to total phytoplankton biomass across the three simulations, with higher biomass across more latitudes in the north compared to the south, although primary production differs from phytoplankton at the equator where it reaches a similar magnitude peak as in the south (Fig. 10). Export production has a markedly different zonal mean distribution across latitudes than PFT biomass and primary production, with the highest production in the tropics for all three simulations. The large variation in zooplankton biomass in the north and south between the three simulations is not reflected in export production, as would be expected (Fig. 10). Around 40ºS and 0º PlankTOM10 primary production peaks and is the highest of the three simulations. This is reflected in PlankTOM10 export peaking at the same latitudes. Around 30-55ºN PlankTOM11 primary production peaks and is the highest of the three simulations, but this is not reflected in PlankTOM11 export peaking over the same latitudes (Fig. 10). Due to the lower total zooplankton biomass in PlankTOM11 compared to the other two simulations, mostly due to the reduced macrozooplankton, driven by the peak in jellyfish biomass. primary production peaks as there is reduced grazing on phytoplankton, but due to lower zooplankton biomass and therefore less zooplankton egestion, excretion and mortality there is less production of $POC_L$.

Globally primary production is higher in PlankTOM10, than in PlankTOM11, but export is slightly lower, as are $POC_S$ and $POC_L$ (Table 6; Fig. A2), indicating that more of the carbon is retained and circulated in the plankton ecosystem in PlankTOM10 than in PlankTOM11. This is not just due to an additional top PFT, as in PlankTOM10.5, primary production and export are the lowest (Table 6; Fig. A2). However, as mentioned previously, the changes to export are smaller than expected given the large changes to zooplankton biomass and ecosystem structure. This is likely due to a bottle neck effect in the model structure, where, for example, mortality from three zooplankton PFTs, enters a single pool (Fig. 1b).

# 4 DISCUSSION

Model results suggest high competition between macrozooplankton (crustaceans) and jellyfish, with a key role of jellyfish being its control on macrozooplankton biomass, which via trophic cascades influences the rest of the plankton ecosystem, across plankton community structure, spatiotemporal dynamics, and biomass. The growth rate of jellyfish is higher than that of macrozooplankton for the majority of the ocean (where the temperature is less than ~25°C) but the mortality of jellyfish is also significantly higher than macrozooplankton, again for the majority of the ocean. The combination of high growth and mortality means that jellyfish have a high turnover rate in temperate waters. In situations where jellyfish mortality is reduced (but still higher than macrozooplankton mortality), jellyfish outcompete macrozooplankton for grazing. Below 20°C jellyfish and macrozooplankton respiration is almost the same, so will have minimal influence on their relative biomass. Biomass is not linearly related to the growth, respiration and mortality rates, with biomass also dependent on prey availability, total PFT biomass and other variables. Because jellyfish also prey directly on macrozooplankton, the biomass of macrozooplankton can rapidly decrease in a positive feedback mechanism. Within oligotrophic regions both jellyfish and macrozooplankton biomass is low, as expected due to limited nutrients limiting phytoplankton growth in these regions. Around equatorial upwelling regions, macrozooplankton outcompete jellyfish.

Macrozooplankton also outcompete jellyfish in many coastal areas including around northern Eurasia because they have a built-in coastal and under-ice advantage to represent enhanced recruitment in these environments which likely tips the balance in their favour (Le Quéré et al., 2016). Around 40°S and 40-50°N jellyfish mostly outcompete macrozooplankton, water temperature here is around 10-17°C which is a temperature were jellyfish growth is the most above macrozooplankton growth and macrozooplankton mortality nearing jellyfish mortality, which combined together favour jellyfish over macrozooplankton. This sensitivity of the composition of the zooplankton community to the mortality of jellyfish could help explain why jellyfish are seen as increasing globally. A reduction in jellyfish mortality during early life-stages i.e. through reduced predation on ephyrae and juveniles by fish (Duarte et al., 2013; Lucas et al., 2012), could quickly allow jellyfish to outcompete other zooplankton, especially macrozooplankton.

The high patchiness of jellyfish in the observations is partly but not fully captured in PlankTOM11 (Fig. 7 and Table 7). The reasons for limited patchiness include the model resolution of ~2°x1° which doesn't allow for the representation of small-scale physical mixing such as eddies and frontal regions, which have been shown to influence bloom formation (Benedetti-Cecchi et al., 2015; Graham et al., 2001). Physical processes are likely to be more responsible for jellyfish patchiness than behaviours, due to their simplistic locomotion. For example, many jellyfish blooms occur around fronts, upwelling regions, tidal and estuarine regions, and shelf-breaks where currents can aggregate and retain organisms (Graham et al., 2001). A few large individuals of the species *Rhizostoma octopus* (barrel jellyfish) have been found to have the capacity to actively swim counter current that could aim to orientate themselves with currents, with the potential to aid bloom formation and retention (Fossette et al., 2015). However, this active swimming behaviour is not representative across the group and would only move the jellyfish within an area less than the resolution of the model. Furthermore, there is currently insufficient data and an incomplete understanding of such swimming behaviours to include it in a global model.

The maximum biomass of jellyfish in PlankTOM11 is 98.9 $\mu$g C L$^{-1}$, compared to the observed maximum biomass of 156 $\mu$g C L$^{-1}$ and the mean:max ratio is within the range of observations although towards the lower end (Table 7). This demonstrates that even without replication of high patchiness, PlankTOM11 still achieved some ephemeral blooms where jellyfish achieved a high biomass.

A key limitation of the representation of jellyfish in the model is the exclusion of the full life cycle. Most jellyfish display metagenesis, alternating between a polyp phase that reproduces asexually and a medusa phase tat reproduces sexually (Lucas and Dawson, 2014). PlankTOM11 currently only characterises the pelagic phase of the jellyfish life cycle, with parameters based on data from the medusae and ephyrae. The biomass of jellyfish is maximal during the pelagic medusa stage, as medusae are generally several orders of magnitude larger than polyps and one polyp can release multiple ephyrae into the water column (Lucas and Dawson, 2014). Although most hydromedusae persist in the plankton for short periods of time, larger scyphomedusae can live for 4-8 months and individuals in some populations can survive for more than a year by over wintering; something that may be facilitated by global climate change (Boero et al., 2016). Polyps develop from planula larvae within 5 weeks of settlement, and can persist far longer than medusae owing to their asexual mode of reproduction and the fact that they can encysts, which allows them to remain dormant until environmental conditions are favourable for budding (Lucas and Dawson, 2014). Unusually, mature medusae of *Turritopsis dohrnii* can revert back to the polyp stage

and repeat the life cycle, which effectively confers immortality (Martell et al., 2016). Our understanding of polyp
ecology is almost entirely based on laboratory reared specimens of common, eurytolerant species, with the
patterns observed being locale- and species-dependent. We know that temperature changes can trigger the budding
of ephyrae by scyphopolyps, which may lead to an increase in the medusa population (Han and Uye, 2010; Lucas
and Dawson, 2014), but the number of species whose polyps have been located and studied in situ is minuscule
and so estimates of polyp abundance or biomass are impossible even to estimate.
Models that include the full jellyfish life cycle are still relatively new, and their focus has been locale- and species-
dependant (e.g. Henschke et al., 2018; Schnedler-Meyer et al., 2018). The aim of this study was not to reproduce
small-scale blooms, but rather to assess at the large and global scale the influence of jellyfish on the plankton
ecosystem and biogeochemistry. We consider it enough to note that higher temperature within PlankTOM11
increases the growth rate, which translates into increased biomass if there is sufficient food, thus providing a
representation of an increasing medusa population. The inclusion of jellyfish life cycles into PlankTOM11 would
introduce huge uncertainties due to the lack of clear in situ life cycle data and is beyond the scope of the exercise.
There is currently no coastal advantage for jellyfish included in the model, as there is for macrozooplankton,
which have a coastal and under-ice advantage for increased recruitment (Le Quéré et al., 2016). Introducing a
similar coastal advantage for jellyfish could introduce an element of life cycle benefits i.e. the increased
recruitment and settlement of planula larvae onto hard substrate in coastal regions and also ephyrae released from
nearshore systems may benefit from being in nearshore waters (restricted there by mobility and current-closure
systems) in much the same way as for other neritic planktonic taxa (Lucas et al., 2012). Alternatively, a deep-
water disadvantage could be introduced for jellyfish to introduce an element of their life cycle dependencies in
that the polyps require benthic substrate for settlement and development into the next life stage and are dependent
on plankton for food, which are more abundant in shallower coastal waters. Future work on PlankTOM11 could
investigate the strengths and weaknesses of these two avenues (coastal advantage and deep-water disadvantage)
for introducing a jellyfish lifecycle element.
Jellyfish in PlankTOM11 are parameterised using data largely from temperate species, because this is the majority
of the data available. This may explain some of the prevalence of jellyfish in PlankTOM11 at mid- to high-
latitudes and the lower biomass in the tropics. Experimental rate data for a wider range of jellyfish species from
a wider range of latitudes is required to address this bias. Another limitation of jellyfish representation in the
model is the lack of body size representation. Generally smaller individuals have greater biological activity, while
larger individuals have greater biomass. Depending on the time of year and life history strategy the dominant
source of biomass will shift between smaller and larger individuals. The size distribution of body mass in jellyfish
is particularly wide compared to other PFTs (Table 1), so representing jellyfish activity by an average sized
individual could well skew the results.
Trophic interactions explain the improvement of spatial chlorophyll with the introduction of jellyfish to the model
(PlankTOM10 to PlankTOM10.5 to PlankTOM11), especially the North/South ratio. The three simulations have
identical physical environments, with the influence of jellyfish as the only alteration, so any differences between
the three can be attributed to the ecosystem structure. Jellyfish are the highest trophic level represented in
PlankTOM11, with preference for meso-, followed by proto-, and then macrozooplankton. However, the largest

influence of jellyfish is on the macrozooplankton, because the grazing pressure on mesozooplankton from macrozooplankton is reduced, and the grazing on protozooplankton by macro- and mesozooplankton is reduced, while the grazing pressure from jellyfish on both meso- and protozooplankton is increased. The combined changes to macrozooplankton and jellyfish grazing pressure counteract to reduce the overall change in grazing pressure. The top down trophic cascade from jellyfish on the other zooplankton also changes some of the grazing pressures on the phytoplankton, which translates into regional and seasonal effects on chlorophyll. Jellyfish increase chlorophyll in the northern pacific and reduce it in the southern pacific, relative to PlankTOM10 (Fig. 9). Seasonally, in the global north jellyfish increase phytoplankton biomass most during the summer and in the global south jellyfish decrease phytoplankton biomass most during the summer, relative to PlankTOM10 (Fig. 11). In the north, most of this summer increase in phytoplankton comes from coccolithophores and *Phaeocystis*, while in the south most of the summer decrease comes from coccolithophores, picophytoplankton and mixed phytoplankton (Fig. A1).

The complexity of zooplankton has been increased, however, the complexity of particulate organic carbon has not, resulting in a bottleneck in carbon export. The low sensitivity of the modelled export to changes in zooplankton composition is likely due to the small number of particulate organic carbon pools. For example, $POC_L$ would export the same carbon particulate whether mesozooplankton, macrozooplankton or jellyfish dominate. There is variety built into the zooplankton contribution to $POC_L$ as the amount entering is dependent on the grazing rate, growth, biomass etc. of each zooplankton, but it all becomes one type of particulate matter once it enters the pool.

The two pools of particulate organic carbon in PlankTOM11 are insufficient to represent the variety of particulate organic carbon generated by the increased variety of zooplankton as the model has been developed. The contribution of mortality to $POC_L$ is orders of magnitude different between mesozooplankton and jellyfish carcases. The composition of the carcases is also very different, with the high water-content of jellyfish compared to other zooplankton, which effects the carcase sinking behaviour (Lebrato et al., 2013a). Mass deposition events of jellyfish carcases (jelly-falls), at depths where the carbon is unlikely to be recycled back into surface waters at short to medium time scales, are known to contain significant amounts of carbon and can contain in excess of a magnitude more carbon than the annual carbon flux (Billett et al., 2006; Yamamoto et al., 2008). PlankTOM11 likely substantially underestimates jellyfish contribution from mortality (Luo et al., 2020). Through rapidly sinking jelly-falls, jellyfish cause a large pulse in export (Lebrato et al., 2012, 2013a, 2013b), not yet accounted for in PlankTOM11. The global export in PlankTOM11 (7.11 PgC/y) is within global estimates of 5 - 12 PgC/y. The main reason for export being towards the lower end of observations is that the global primary production in PlankTOM11 is lower than the observed rate. Another potential explanation which may enhance the low export is that within the model jellyfish have a high turnover rate, due to their high growth, grazing and mortality rates, thus taking in a high proportion of carbon, but they are not then acting as a direct rapid source of sinking carbon through their mortality.

 The contribution of egestion and excretion (see Fig. 1b and Fig. A2) to $POC_L$ is also very different between mesozooplankton, macrozooplankton and jellyfish, most particularly that the main contribution from meso- and macrozooplankton is in the form of solid faecal pellets, while for jellyfish the main contribution is from mucus (Hansson and Norrman, 1995). The composition and sinking behaviour of faecal pellets and mucus will be

substantially different, with mucus sinking more slowly and more likely to act as a nucleus for enhanced aggregation with other particles, forming a large low-density mass (Condon et al., 2011; Pitt et al., 2009).

Work is currently underway on PlankTOM to increase the size partitioning of particulate organic carbon through introducing a size-resolving spectral model with a spectrum of particle size and size-dependent sinking velocity (Kriest and Oschlies, 2008). This method has the advantage of improving the representation of particulate organic carbon production from all PFTs but is substantially more computer expensive. Another role of jellyfish may be that they act as significant vectors for carbon export, but with the current POC partitioning in PlankTOM11 this role has not been elucidated here. The potential influence of introducing increased size partitioning on carbon export could be significant, with peaks in jellyfish biomass being followed by a pulse in carbon export as there is rapid sinking of large carcasses (Lebrato et al., 2012; Luo et al., 2020).

Jellyfish have been included in a range of regional models, the majority are fisheries-based ecosystem models, namely ECOPATH and ECOPATH with ECOSIM (Pauly et al., 2009). These include regional models of the Northern Humboldt Current system (Chiaverano et al., 2018), the Benguela Upwelling System (Roux et al., 2013; Roux and Shannon, 2004; Shannon et al., 2009) and an end-to-end model of the Northern California Current system, based on ECOPATH (Ruzicka et al., 2012). Jellyfish have also been included in regional Nutrient Phytoplankton Zooplankton Detritus (NPZD) models, representing small-scale coastal temperate ecosystems with simple communities, for example, Schnedler-Meyer et al. (2018) and Ramirez-Romero et al. (2018). These models have provided valuable insight into jellyfish in the regions studied, but the focus on coastal ecosystems and either a top-down approach (ECOPATH) or highly simplified ecosystem (NPZD) limits their scope. A recent paper has included jellyfish in a global ecosystem model, including multiple other zooplankton and fish types and provides a static representation of biomass (Heneghan et al., 2020). However, the model does not include phytoplankton, biogeochemistry (outside of using carbon content to determine zooplankton functional groups) or any ocean physics. PlankTOM11 offers the first insight into the role of jellyfish on plankton community structure, spatiotemporal dynamics, and biomass, using a global biogeochemical model that represents multiple plankton functional types.

## 3.5 CONCLUSION

Jellyfish have been included as a PFT in a global ocean biogeochemical model for the first time as far as we can tell at the time of writing. The PlankTOM11 model provides reasonable overall replication of global ecosystem properties and improved surface chlorophyll, particularly the north/south ratio. The replication of global mean jellyfish biomass, 0.13 PgC, is within the observational range, and in the region with the highest density of observations PlankTOM11 closely replicates the mean and seasonal jellyfish biomass. There is a deficit of data on jellyfish carbon biomass observations and physiological rates. Monitoring and data collection efforts have increased over recent years; we recommend a further increase especially focussing in less-surveyed regions and on non-temperate species.

The central role of jellyfish is to exert control over the other zooplankton, with the greatest influence on
macrozooplankton. Through trophic cascade mechanisms jellyfish also influence the biomass and spatiotemporal
distribution of phytoplankton. PlankTOM11 is a successful first step in the inclusion of jellyfish in global ocean
biogeochemical modelling. The model raises interesting questions about the sensitivity of the zooplankton
community to changes in jellyfish mortality and calls for a further investigation in interactions between
macrozooplankton and jellyfish. Future model development, alongside POC improvements, could include an
exploration of the life cycle, coastal advantages, and higher resolution ocean physical processes to enhance
patchiness.

# Appendix

**Table A1:** Sources and metadata for jellyfish growth rates, including references with associated number of data points, species and life stage used to inform the growth parameter of jellyfish in PlankTOM11.

| Reference | $n$ | Species | Life Stage |
|---|---|---|---|
| Båmstedt et al., (1997) | 3 | *Cynea capillata* | Ephyrae |
| Daan (1986) | 8 | *Sarsia tubulosa* | Medusae |
| Frandsen & Riisgård (1997) | 5 | *Aurelia aurita* | Medusae |
| Hansson (1997) | 20 | *Aurelia aurita* | Medusae |
| Møller & Riisgård (2007a) | 34 | *Sarsia tubulosa, Aurelia aurita, Aequorea vitrina* | Medusae, ephyrae |
| Møller & Riisgård (2007b) | 10 | *Aurelia aurita* | Medusae, ephyrae |
| Olesen (1994) | 8 | *Aurelia aurita, Chrysaora quinquecirrha* | Medusae, ephyrae |
| Widmer (2005) | 10 | *Aurelia labiata* | Ephyrae |

**Table A2:** The fit to the growth data for PFT's for the new three-parameter fit used in this study (see Eq. 3 and Fig. 2) and the two-parameter fit (see Eq. 2 and Fig. 2).

| PFT | $R^2$ | | $n$ |
|---|---|---|---|
| | Two-parameter | Three-parameter | |
| CNI | 9.58 | 11.36 | 98 |
| MAC | 36.57 | 36.76 | 253 |
| MES | 0.32 | 0.34 | 2742 |
| PRO | 0.00 | 7.81 | 1300 |
| BAC | 1.66 | 1.66 | 1429 |
| DIA | 9.59 | 9.58 | 439 |
| PHA | 6.29 | 37.07 | 67 |
| MIX | 21.25 | 19.17 | 95 |
| COC | 33.91 | 36.01 | 322 |
| PIC | 20.17 | 20.29 | 150 |
| FIX | 2.67 | 10.62 | 32 |

**Table A3:** Sources and metadata for jellyfish grazing preferences, including references with associated species, life stage and preference for prey (categorised into PFTs) with any notable phrases used to inform the grazing of jellyfish in PlankTOM11.

| Reference | Species/Class/Genera | Life Stage | PFT preference |
|---|---|---|---|
| Båmstedt et al. (2001) | *Aurelia aurita* | Ephyrae | Mixed-phytoplankton, mesozooplankton and particulate organic material |
| Colin et al. (2005) | *Aglaura hemistoma* | Medusa | "microplanktontic omnivores"; protozooplankton and some phytoplankton |
| Flynn and Gibbons (2007) | *Chrysaora hysoscella* | Medusa | Wide variety ranging in size from protozooplankton to macrozooplankton, with the "numerically dominant" prey as mesozooplankton |
| Malej et al. (2007) | *Aurelia* sp. | Medusa | Mesozooplankton and protozooplankton |
| Morais et al. (2015) | *Blackfordia virginica* | Medusa | Mesozooplankton and diatoms |
| Purcell (1992) | *Chrysaora quinquecirrha* | Medusa | Mesozooplankton (upto 71% of diet) |
| Purcell (1997) | Hydromedusa | | "mostly generalist feeders", mesozooplankton as a preference |
| Purcell (2003) | *Aurelia labiata, Cyanea capillata, Aequorea aequorea* | | Mainly mesozooplankton |
| Stoecker et al. (1987) | *Aurelia aurita* | Medusa | Protozooplankton and mesozooplankton preferentially removed from "natural mircozooplankton" assemblage. In cultured prey assemblage, larger protozooplankton were selected. |
| Uye and Shimauchi (2005b) | *Aurelia aurita* | Medusa | Mostly mesozooplankton, some protozooplankton |
| Costello and Colin (2002) | *Aglantha digitale, Sarsia tubulosa, Proboscidactyla flavicirrata, Aequorea victoria, Mitrocoma cellularia, Phialidium gregarium* | Medusa | Mesozooplankton (crustacean) and protozooplankton (ciliates) |







**Table A4:** Additional tuning parameter values for PlankTOM11 (see Sect.2.1.5) following the change to the growth rate formulation. 'Before growth change' values are those used in PlankTOM10[LQ16] and 'after growth change' values are used in simulations for this study (PlankTOM11, PlankTOM10.5 and PlankTOM10).

| Parameter | Before growth change | After growth change |
|---|---|---|
| Grazing preference ratio of mesozooplankton for *Phaeocystis* | 0.75 | 1 |
| Grazing preference ratio of protozooplankton for picophytoplankton | 2 | 3 |
| Half saturation constant of phytoplankton grazing on iron | | |
| Diatoms | 40.0e-9 | 80.0e-9 |
| Picophytoplankton | 10.0e-9 | 25.0e-9 |
| *Phaeocystis* | 25.0e-9 | 80.0e-9 |
| Half saturation constant of bacteria for dissolved organic carbon | 10.0e-6 | 8.0e-7 |
| Maximum bacteria uptake rate | 3.15 | 1.90 |
| Diatom respiration | 0.012 | 0.12 |














**Table A5.** Global mean values for rates and biomass from observations with the associated references. In parenthesis is the percentage share of the plankton type of the total Phytoplankton or Zooplankton biomass.

|  | Observations | Reference for the data |
|---|---|---|
| **Rates** |  |  |
| Primary production (PgC y$^{-1}$) | 51-65 | Buitenhuis et al. (2013b) |
| Export production at 100m (PgC y$^{-1}$) | 5-13 | Henson et al. (2011), Palevsky et al. (2018) |
| CaCO$_3$ export at 100m (PgC y$^{-1}$) | 0.6-1.1 | Lee (2001), Sarmiento et al. (2002) |
| N$_2$ fixation (TgN y$^{-1}$) | 60-200 | Gruber (2008) |
| **Phytoplankton biomass 0-200m (PgC)** |  |  |
| N$_2$-fixers | 0.008-0.12 (2-8%) | Luo et al. (2012) |
| Picophytoplankton | 0.28-0.52 (35-68%) | Buitenhuis et al. (2012b) |
| Coccolithophores | 0.001-0.032 (0.2-2%) | O'Brien et al. (2013) |
| Mixed-phytoplankton | - | - |
| *Phaeocystis* | 0.11-0.69 (27-46%) | Vogt et al. (2012) |
| Diatoms | 0.013-0.75 (3-50%) | Leblanc et al. (2012) |
| **Heterotrophs biomass 0-200m (PgC)** |  |  |
| Bacteria | 0.25-0.26 | Buitenhuis et al. (2012a) |
| Protozooplankton | 0.10-0.37 (27-31%) | Buitenhuis et al. (2010) |
| Mesozooplankton | 0.21-0.34 (25-66%) | Moriarty and O'Brien (2013) |
| Macrozooplankton | 0.01-0.64 (3-47%) | Moriarty et al. (2013) |
| Jellyfish zooplankton | 0.10-3.11 | Bar-On et al. (2018), Lucas et al. (2014), Buitenhuis et al. (2013b) |






**Table A6:** Total phytoplankton biomass ($\mu$mol C L$^{-1}$) for 30°N – 70°N across all longitudes. Observations are from gridded MAREDAT, all data are for the surface ocean (0-10 meters). Phytoplankton types include picophytoplankton, *Phaeocystis*, diatoms, nitrogen-fixers and coccolithophores. The seasonal amplitude is the amplitude for the full seasonal cycle (January – December) and the non-winter amplitude is the amplitude for March – October.

|  | Seasonal Amplitude | Non-winter Amplitude |
|---|---|---|
| Observations (median – mean) | 0.78 – 2.67 | 0.70 – 2.12 |
| PlankTOM11 | 1.82 | 0.97 |
| PlankTOM10.5 | 1.54 | 0.80 |
| PlankTOM10 | 1.69 | 0.81 |
| PlankTOM10$^{LQ16}$ | 1.68 | 1.02 |



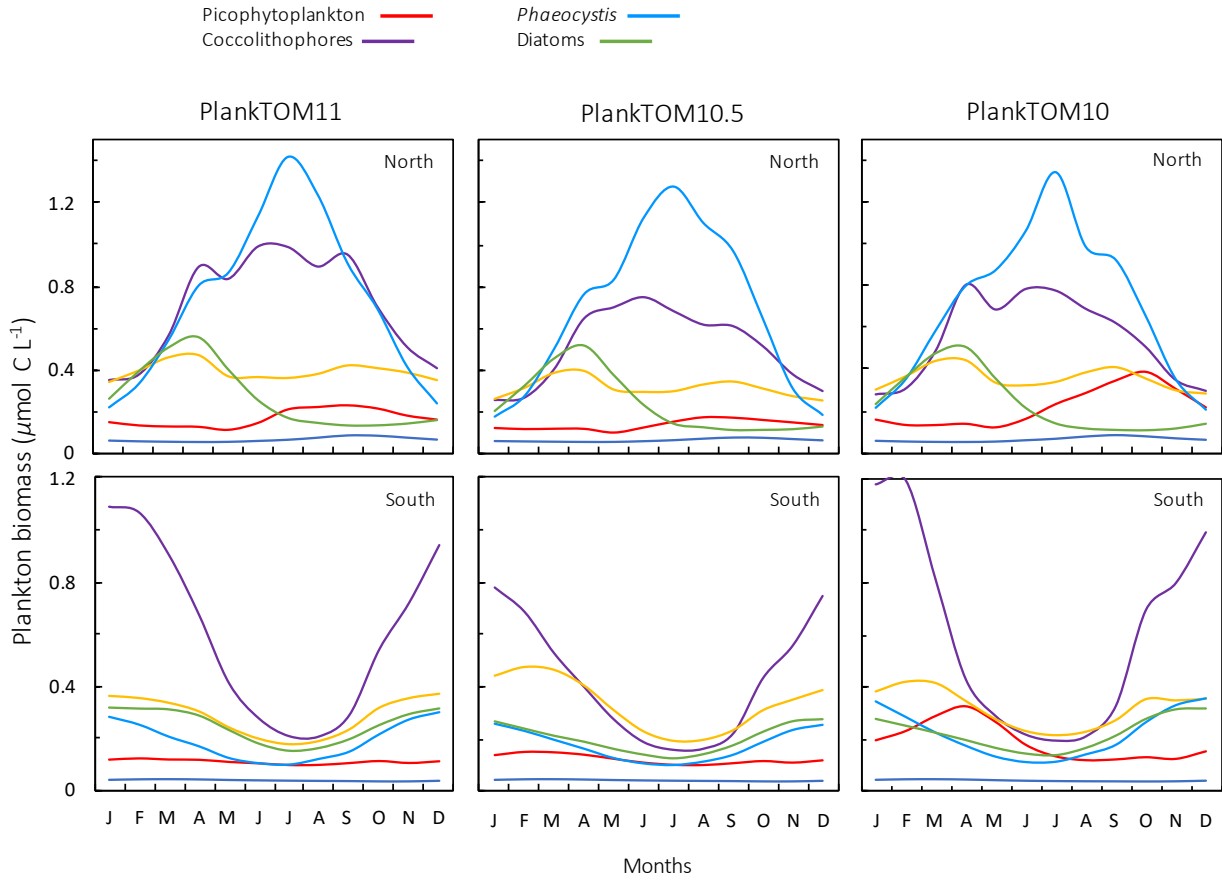


***Figure A1.*** *Seasonal surface carbon biomass (μmol C L$^{-1}$) of phytoplankton PFTs; N$_2$ fixers, picophytoplankton,*
*coccolithophores, mixed phytoplankton, Phaeocystis and diatoms. Panels shown PFT biomass for PlankTOM11 (left),*
*PlankTOM10.5 (middle) and PlankTOM10 (right), for two regions; the north 30ºN - 70ºN (top) and the south 30ºS - 70ºS*
*(bottom) across all longitudes. All data are averaged for 1985-2015.*

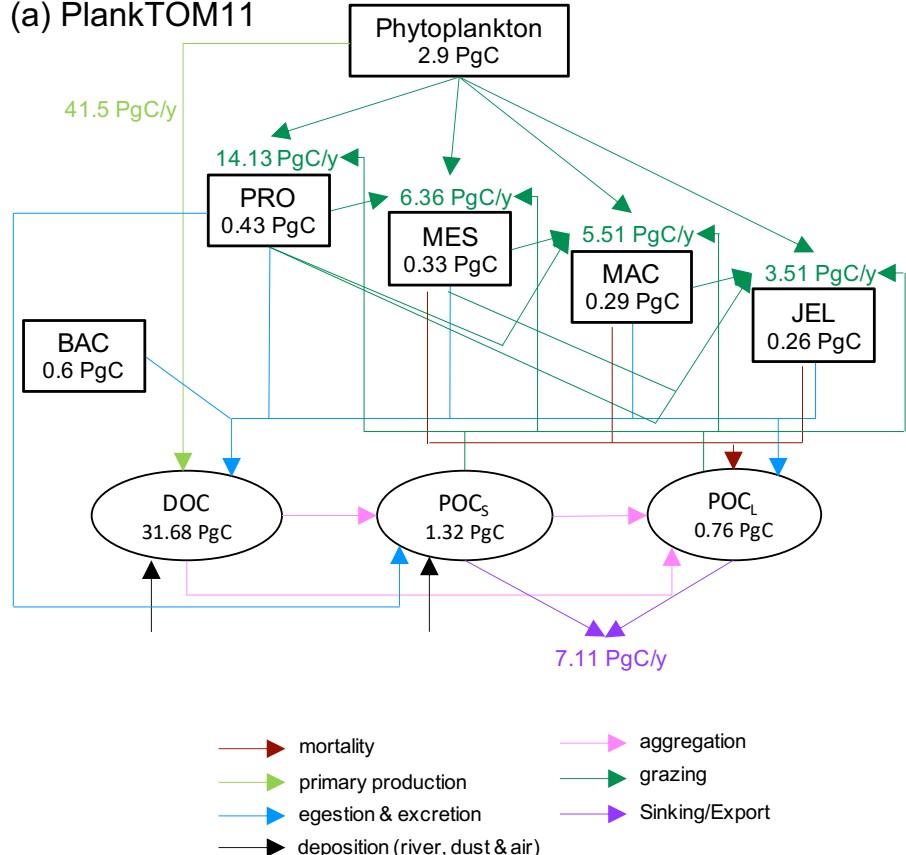


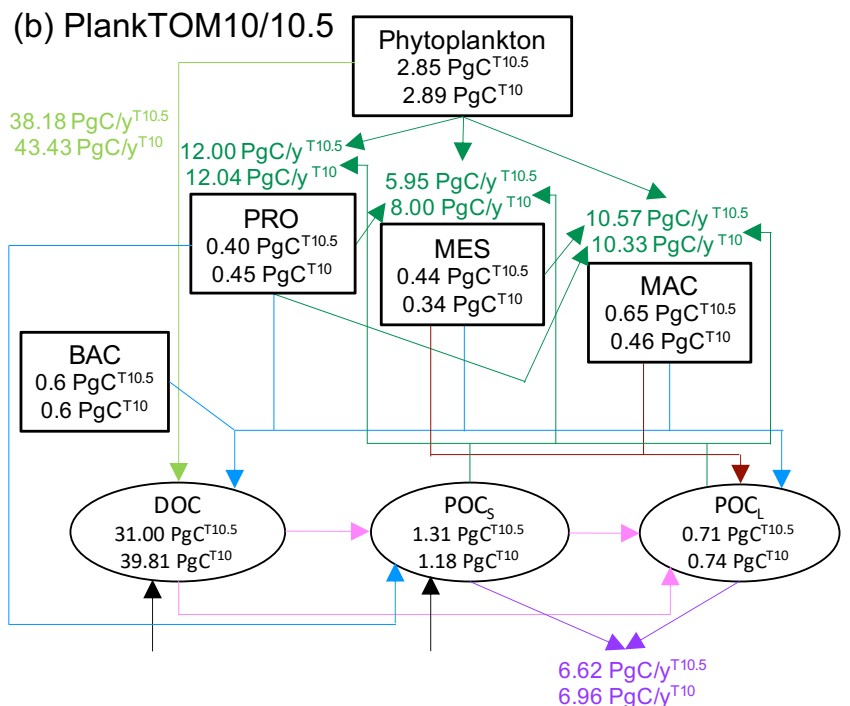


*Figure A2. Schematic representation of global carbon biomass and rates in the PlankTOM marine ecosystem model including sources and sinks for dissolved organic carbon (DOC) and small (POCS) and large (POCL) particulate organic carbon. (a) PlankTOM11 and (b) PlankTOM10 and PlankTOM10.5. Carbon biomass (PgC) of PFT's and organic carbon pools are given within boxes and ovals, carbon rates (PgC/y) of primary production (light green), grazing (dark green) and export production (purple) are given next to the corresponding arrows. All data are averaged for 1985 to 2015.*

## Author Contribution


RMW, CLQ, ETB and SP conceptualized the research goals and aims. RMW carried out the formal analysis
with contributions from CLQ and ETB. RW developed the model code with significant contributions from ETB,
and RMW performed the simulations. RMW prepared the manuscript with contributions from all co-authors.
The authors declare that they have no conflict of interest.

## Acknowledgements


RMW was funded by Doctoral Training Programme ARIES, funded by the UK Natural Environment Research
Council (project no. NE/L002582/1). CLQ was funded by the Royal Society (grant no. RP\R1\191063). ETB
was funded by the European Commission H2020 project CRESCENDO (grant no. 641816). This research was
partly conducted in South Africa with the support of the Newton International PhD exchange programme (grant
no. ES/N013948/1). The model simulations were done on the UEA's High Performance Computing Cluster.

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
