# Peer review of "Role of jellyfish in the plankton ecosystem revealed using a"

_Biogeosciences, 2020_

## Referee Comment (RC1) · Anonymous Referee #1 · 23 Jun 2020

**General Comments**

The paper addressed relevant scientific questions within the scope of Biogeosciences by examining marine biogeochemistry and planktonic ecosystem function. It presents a novel global model of ocean physics and biogeochemistry that adds a jellyfish plankton functional type (PFT) to the typically represented phytoplankton and crustacean zooplankton. Additionally, its use of 11 PFTs resolves the planktonic ecosystem more finely than most coupled physical-biogeochemical ocean models. The scientific methods and assumptions of the paper are valid but need to be more clearly outlined. Specifically, all new parameter values must be given to allow reproduction (traceability

of results). The results are sufficient to support the authors' interpretations and conclusions, but neither the results nor their interpretations/conclusions are very substantial. Similarly, the Tables and Figures illustrate the results well, but lack some of the detailed methodology and analysis. The title reflects the contents of the paper, though "unique" may not be the most accurate word and requires substantiation. Overall, the paper is well-structured with a clear flow. With a deeper analysis and discussion of mechanisms, this paper would be an unparalleled contribution ocean biogeochemical modeling.

**Specific Comments**

1.

The words "unique" and "role" are vague in the title and not well defined throughout the paper.

1a. "Unique" may not be the most accurate word. Other organisms could play a similar ecological role as jellyfish as predators on and competitors with macrozooplankton. For example, fish larvae, squids, and benthic filter feeders. If the authors wish to use this term they should substantiate the uniqueness of the jellyfish role in the Discussion. I have the same issue with its use in the abstract.

1b. L91-92: Vague what is meant by the "role of jellyfish." What quantitative metrics are used to assess this?

2.

Lacking sufficient methodological detail for reproducibility.

2a.  L86-89: If the macrozooplankton only represent crustaceans here, then they should not eat picoplankton. The salps and pteropods that were included in the group as described in Le Quéré et al. (2016) can do this, but euphausiids do not.

2b. L103-104: This is not true; many parameters have been modified. e.g. L260-266.

2c.  L129-143: Please mention here that the $g_{F_k}^{Z_j}$ term is a temperature-dependent Michaelis-Menten form that includes the prey preference and a half saturation coefficient. Otherwise, please provide the full equations.

2d. Eq 1: The authors mention in the Introduction that jellyfish are part of the biological pump and may be a significant vector for carbon export. How does the jellyfish PFT contribute to the 3 detrital pools? Are the parameters the same as or different than the other zooplankton groups?

2e. L165-166: What is the GGE value for the jellyfish?

2f.  L181-183 and Table 3: The preferences are a ratio of what to what?  What are the numerator and denominator?  It is unclear how preference enters into the grazing equation and how preferences can be greater than one.  I had to refer back to the Le Quéré et al. 2016 paper (LQ16).  I think that this information is essential and the reader should not have to look at the original paper.

2g. L259-260: These new parameters should be given for transparency and repro-ducibility. A table in the Appendix would be acceptable.

2h. L279-290: This description lacks some necessary details. For example, it says that PlankTOM10 is the same as PlankTOM11 except that the top predator mortality terms for mesozooplankton and macrozooplankton were returned to pre-jellyfish values. One, I assume that pre-jellyfish values are those from Le Quéré et al. 2016 (but Table 5 does not support this). Two, did the mesozooplankton respiration rate also return to pre-jellyfish values? (Yes, as evidenced in Table 5, but not mentioned in the text) Three, all of the changes to phytoplankton and bacteria made to PlankTOM11 remained? Does PlankTOM10.5 just have two identical macrozooplankton groups? I would suggest a table that outlines the differences between all 3 simulations. The authors attempted to do this with Table 5, but it lacks a column for PlankTOM10.5 and there are discrepancies between the text here and the values in the table. If PlankTOM10 mortality rates were returned to pre-jellyfish values, then why are there different MAC mortality rates for PlankTOM10 (LQ16) and PlankTOM10 (this study) in the table? And are macrozooplankton mortality and mesozooplankton respiration the only 2 parameters that varied across the 3 simulations?

2i. Table 4: Please show all parameters for jellyfish and macrozooplankton. e.g. grazing rate temperature-dependence, half saturation coefficients, MGE, GGE, etc.

2j. Table 6: What is adapted from Le Quéré et al. 2016? Is the PlankTOM10 in this table PlankTOM10-LQ16 or PlankTOM10-here? That needs to be noted in the table caption.

2k. Figure 8: The black boxes that denote the North, Tropic, and South regions used in Figures 4 and 9 are only in the Pacific. If this analysis was only carried out for the Pacific, then that should be mentioned in the main text as well.

3.

The analysis is rather superficial. It only described changes in plankton biomasses. I expected Section 3.3 to also compare differences in net primary production, carbon export, nutrient cycling, etc. with and without jellyfish. More detail should be given on how rates and fluxes change across the three simulations as well as on the mechanisms and processes involved. e.g. What are the grazing mortality rates of each PFT by each predator with and without jellyfish? How does the productivity of each PFT change? What are the flows of mass/energy from one PFT to another in the three simulations? These types of analyses would better elucidate food web structure and function with and without jellyfish.

3a. L421-422: Why merely suggest that the decrease in predation of protozooplankton by macrozooplankton may be compensated for by the increase in predation by jellyfish? Why not use the model output to verify this? This is one example of how this manuscript would be improved by a more thorough analysis. For example, the mortality of each PFT could be partitioned by each grazer in the 3 simulations.

3b. L389-395: The partitioning of phytoplankton biomass by PFT differs between the PlankTOM11 and observations. Is the PlankTOM10 in Table 6 PlankTOM10-LQ16 or PlankTOM10-here? Was the partitioning of different phytoplankton PFTs in PlankTOM10-LQ16 the same as here with PlankTOM11? Or did the partitioning change? If it stayed the same, it suggests that the jellyfish had no effect on phytonone

plankton community composition. If it changed, did it become more or less aligned with observations and how did the jellyfish affect it?

3c. L413-414: It is very unclear why the biomass of macrozooplankton drops from PlankTOM10-here to PlankTOM10.5 and this needs to be described in further detail. As far as I can tell, PlankTOM10-here and PlankTOM10.5 are nearly identical, except that macrozooplankton mortality and mesozooplankton respiration are lower in PlankTOM10.5 and there are two identical macrozooplankton groups. Without any parameter changes, I expect the sum of the two macrozooplankton groups in PlankTOM10.5 to equal the biomass of the macrozooplankton in PlankTOM10-here. But the decrease in macrozooplankton mortality would lead me to expect an increase in macrozooplankton biomass in PlankTOM10.5. Why does it decrease? Did the drop in mesozooplankton respiration allow them to outcompete macrozooplankton for shared resources? And what accounts for the change in latitudinal distribution of the mesozooplankton and macrozooplankton in PlankTOM10.5?

3d. The Discussion does a thorough job addressing the assumptions and limitations of the model. However, it is lacking a section that describes the hypothesized mechanisms involved in the differences between PlankTOM10-LQ16, PlankTOM10-here, PlankTOM10.5, and PlankTOM11.

3e. The Discussion also lacks a section on how the lower temperature sensitivity of jellyfish (lower $Q_{10}$) compared to macrozooplankton might affect spatial distributions and how this is balanced or offset by disparities in their respiration and mortality rates.

3f. L509-511: More detail should be added here to describe where regionally, when seasonally, and which phytoplankton PFTs.

4.

The model really only characterizes the pelagic phase of the complex jellyfish life cycle. The authors cited much variability in this life cycle, but do not provide enough information on how representative this model is without the life cycle and dependence on benthic substrate? Some useful details include how much time is spent the pelagic medusa stage and how much biomass is present in this stage in comparison to the other stages.

Other scientific questions/issues:

L152-155: Please add a statistical skill metric for the exponential fit and the 3-parameter fit to observations so that the reader may compare. Showing both in Figure 2 for the jellyfish could also help support the claim that the exponential fit is poor.

L228: How was the adjusted mortality of $\mu = 0.12$ chosen from the sensitivity analysis? What skill metric was used?

L251-254: Jellyfish had a higher preference for protists than microzooplankton. Why were changes unnecessary for the protist parameters?

L272: Why was 1948 not used for spin-up, since this would be the start year of meteorological forcing. Couldn't using 1980 to spin-up induce a shock to the system at 1948 that would then need to stabilize?

L307-308: Why were the MAREDAT observations binned to a different grid as the model? Why not use the same grid?

L338: Please add a global map of observations for visual comparison.

L348-349: Why not use the same type of mean to compare? Or if the authors are concerned about the underrepresentation of zeros in the observations, why not use the mean that is best for describing that type of distribution?

L352-361: This paragraph is missing a sentence that notes where the model disagrees with observations spatially. There is a prominent difference out the outer shelf of the Eastern Bering Sea where the model predicts some of the highest biomasses while observations show some of the lowest biomasses. A potential explanation for this discrepancy should also be added here or in the Discussion as appropriate. (Here if it is local to Alaska, in the Discussion if it is applicable to the model as a whole.)

L376-379: Is this ratio standard for validating model chlorophyll? i.e. Is it a meaningful metric?

L387: This underestimate of primary production by 10 PgC/y seems rather large. How does it compare to the Le Quéré et al. 2016 model and other biogeochemical models?

L395-398: These statements could be supported by mentioning that the light affinity and nutrient uptake parameters of mixed phytoplankton and Coccolithophores are very similar to those of picophytoplankton, with the exception of Fe uptake.

L406-407: But jellyfish have a much higher preferences for mesozooplankton and protozooplankton than macrozooplankton. How does this affect the results?

L432-433: This line is too vague. What was the largest direct influence of jellyfish? Predation? Competition? If competition, for which resources?

L434-437: Is a double peak in northern hemisphere phytoplankton seasonal biomass consistent with observations? Is the amplification more or less similar to observations? Is one of the simulations (PlankTOM10-LQ16, PlankTOM10-here, PlankTOM10.5, PlankTOM11) more similar to observations?

L486-488: The jellyfish may not need a coastal advantage, but a deep-water disadvantage, since their benthic polyps are filter feeders and dependent on pelagic plankton.

Figure 2: Using the same y-axis scale for all subplots hides the fit with observations for FIX, MAC, and JEL. Also, an $R^2$ and/or p-value for the fit would be appreciated.

Figure 4: Could observations of the observed PFT biomasses from MAREDAT be added to this plot similar to observed chlorophyll?

Figures 5, 6, 7, 8: I recommend a colormap that is perceptually ordered for spatial distributions. See cmocean, colorbrewer, and colormoves for examples.

Figure 7: The cyan color used in the time series is very difficult to see. Use a darker color or a dashed black line.

Figures 10, 11: I would not refer to the 2nd macrozooplankton group of PlankTOM10.5 as jellyfish in these figures. Instead, the biomass of the 2 macrozooplankton groups should be summed together and displayed that way. Keeping them distinct is misrepresentative.

---

## Referee Comment (RC2) · Anonymous Referee #2 · 8 Jul 2020

General Comments

The manuscript fits perfectly in the scope of Biogeosciences in that it considers the role of plankton within marine biogeochemistry. It uses a currently developed model PlankTOM10 and adds a jellyfish plankton functional traits (PFT) to further resolve the global ocean plankton system. Gelatinous zooplankton within the Cnidaria and Ctenophora have been neglected in virtually all models, yet we know that they have the potential to play a significant role in structuring plankton food webs directly and indirectly via predation, and facilitate the flux of organic carbon to the seafloor via the production of mucous, messy feeding, carcasses (also known as jelly falls) and to a

much lesser extent faecal pellets. The pelagic tunicates (salps, doliolids) can form substantive bloom events and have the capacity to graze particles down to only a few microns, but these have also not been included.

Although the authors appear to have used appropriate methods and scientific assumptions, it is difficult to make firm judgements about this as there not a detailed justification of the parameter values used. I would expect in a paper such as this for a detailed summary of all sources of parameter inputs listed in the Appendix for readers to check themselves and also reproduce the models. Without this a reader has to go on a fact finding mission themselves. The interpretation of the results are sound but not particularly substantive. The inclusion of jellyfish in the PlankTOM model is a significant step forward but the authors have not really explored this as much as I would have hoped. When I first read the title of the paper I expected a model of carbon flows from one PFT to another presented rather than just biomass outputs, and this left me feeling somewhat disappointed. If these concerns were addressed it would make the work far more powerful and novel.

Specific Comments

L33 - I am not sure jellyfish play a unique role. They do play a role, as do all the other functional groups. What are you suggesting by the term unique?

L34 - There have been a (very few) instances where jellyfish have been considered in plankton ecosystems, e.g., Ruzicka et al 2012 for the California Current System.

L86 - You do not mention, or make clear, what the composition of the macrozooplankton group is. Does it include pelagic tunicates, which are going to graze down to a smaller food size than crustacean macrozooplankton such as euphausiids. This would make a difference to how your model runs.

p114 - What are the size definitions of the two particulate detrital pools? The terms small and large are vague. You do not specify the contribution of each of the PFTs to

each pool. For example, jellyfish produce virtually nothing in the way of particulate / solid faecal pellets.

p135 - What is MGE?

p147 - You mention that jellyfsh growth rates were compiled as a function of temperature from the literature, but you do not provide any indication of which papers were used, how many were used, which taxa the growth rates were compiled from etc. It is this level of detail that is absent from the methodology which makes traceability of the data impossible to verify.

p165-166 - Continuing with the issues of transparency, the values for GGE are obtained from the literature (Moriarty, 2009), but this is difficult to verify a that is a PhD thesis. You should make it clear that the data from the literature have been collated by Moriarty, 2009. How many values were collated? What are the range of values? Stating these will make readers far more confident about the inputs into the model.

p178 - Do you have evidence that ephyrae do have a higher clearance rate for autotrophs. There are not many papers that have analysed diet of ephyrae and there are mixed messages about diet. For example, how can you take into account selective vs non-selective feeders and time of year (relative to the spring phytoplankton blooms)?

L299 - It would be useful to include a map showing the global distribution of jellyfish for the reader to gain a better understanding of the spatial distribution and coverage. Are the Cnidaria data used from the upper 200m only, as you indicate that in the original dataset jellyfish were available for a much wider depth range. Again, this is for transparency purposes.

L319 and 327 - Why do you express the values for jellyfish biomass as 0.46 - 3.11 pg C on line 319 in the methods and just 0.46 pg C on line 327 at the start of the results (where other published values are expressed as a range)?

L342-344 - It is obvious that the majority of jellyfish biomass is distributed around

the coasts because a) that is where the majority of sampling has taken place and so there will be sampling bias, and b) it is likely that the majority of jellyfish collected are scyphomedusae with a metagenic life cycle requiring a hard substrate for the benthic polyp population.

L417-418 - You state there is a high preference for jellyfish on protozooplankton. The vast majority of diet and feeding studies on jellyfish suggest that mesozooplankton are the preferred prey for the majority of jellyfish. Smaller taxa and juvenile forms (ephyrae) would consume protozooplankton, but this is not the case for most of the scyphomedusae. In the jellyfish dataset used for the PlankTOM11 model, are the classes or genera listed? If so, it would be helpful to briefly indicate the make up of the jellyfish community used in this study.

L453 - The grid resolution is stated as ~2o x 1o, but the original dataset were gridded in 1 x 1 degree. Why was the resolution changed?

L472 - Brotz et al. 2012 is not the most appropriate reference to support the description of jellyfish reproduction alternating between a sexually-reproducing pelagic medusa and asexually-reproducing benthic polyp, as the Brotz paper is about global distributions of gelatinous zooplankton and not reproduction.

L472-479 - Be careful about saying that increasing temperatures increase growth of jellyfish, which they do, but it is an oversimplification of the whole life cycle as ephyrae are typically produced following colder than average winter temperatures (certainly for temperate populations of the common jellyfish Aurelia, which likely dominates datasets).

Overall the discussion is rather brief and does not fully explore the differences between the different model outputs and the mechanisms driving those differences. The discussion feels rather superficial and far more explanation is needed to make it robust.

Technical Corrections

L55 - benthic polyp (delete s)

L177 - Obelia in italics

L195 - Aurelia in italics

L197 - data were (not was)

L394 - picophytoplankton (lower case p)

Table 1 - italicise genus names

---

## Author Comment (AC1) · 7 Oct 2020

We sincerely thank the reviewer for their insightful comments, which have greatly helped to improve and clarify our manuscript. We respond in detail to each comment in the attached document, and clarified the points raised by the reviewers. In response to these comments we have implemented the following general changes:

- All parameter values for the jellyfish PFT have been added, with the expansion of tables in the main manuscript and the addition of tables in the appendices, along with further descriptions of the model structure and additional equations.
- Carbon and primary production have been added to the results section and discussion, along with diagrams of the carbon flow within the model.
- The discussion has been expanded throughout to about twice its original length.

We provide full details in the attached, with our replies to reviewers' comments in blue, and the new text provided in the manuscript highlighted in red.

Rebecca Wright, on behalf of the author team.

**Response to the comments**

**Reviewer #1**

General Comments

The paper addressed relevant scientific questions within the scope of Biogeosciences by examining marine biogeochemistry and planktonic ecosystem function. It presents a novel global model of ocean physics and biogeochemistry that adds a jellyfish plankton functional type (PFT) to the typically represented phytoplankton and crustacean zooplankton. Additionally, its use of 11 PFTs resolves the planktonic ecosystem more finely than most coupled physical-biogeochemical ocean models. The scientific methods and assumptions of the paper are valid but need to be more clearly outlined. Specifically, all new parameter values must be given to allow reproduction (traceability of results). The results are sufficient to support the authors' interpretations and conclusions, but neither the results nor their interpretations/conclusions are very substantial. Similarly, the Tables and Figures illustrate the results well, but lack some of the detailed methodology and analysis. The title reflects the contents of the paper, though "unique" may not be the most accurate word and requires substantiation. Overall, the paper is well-structured with a clear flow. With a deeper analysis and discussion of mechanisms, this paper would be an unparalleled contribution ocean biogeochemical modeling.

Thank you for your detailed comments and suggestions. All new parameter values have been added to the paper, either within the text, in tables or in appendix tables. Further analysis, especially in relation to carbon flow, has been added with the discussion points expanded on throughout. See below for specific details.

Specific Comments

**1.** The words "unique" and "role" are vague in the title and not well defined throughout the paper.
**1a.** "Unique" may not be the most accurate word. Other organisms could play a similar ecological role as jellyfish as predators on and competitors with macrozooplankton. For example, fish larvae,

squids, and benthic filter feeders. If the authors wish to use this term they should substantiate the uniqueness of the jellyfish role in the Discussion. I have the same issue with its use in the abstract.

Yes indeed. Unique has been removed from the title and abstract.

**1b. L91-92:** Vague what is meant by the "role of jellyfish." What quantitative metrics are used to assess this?

The role of jellyfish refers to the influence they have on the plankton ecosystem, which is addressed in this study through the comparisons between the three model simulations with modified ecosystem configurations. The key role found in this study is the control jellyfish have over macrozooplankton biomass, which then influences the rest of the plankton ecosystem and the spatial concentration of chlorophyll through trophic cascade mechanisms. Specific points highlighting what is meant by the role of jellyfish have been added.

In section 3.3;

"The largest direct influence of jellyfish in these regions is its role in controlling macrozooplankton biomass, through competition for prey resources, particularly mesozooplankton and protozooplankton, and through the predation of jellyfish on macrozooplankton."

In the discussion;

"Model results suggest high competition between macrozooplankton (crustaceans) and jellyfish, with a key role of jellyfish being its control on macrozooplankton biomass, which via trophic cascades influences the rest of the plankton ecosystem."

In the conclusion;

"The central role of jellyfish is to exert control over the other zooplankton, with the greatest influence on macrozooplankton. Through trophic cascade mechanisms jellyfish also influence the biomass and spatial distribution of phytoplankton."

**2.** Lacking sufficient methodological detail for reproducibility.

**2a. L86-89:** If the macrozooplankton only represent crustaceans here, then they should not eat picoplankton. The salps and pteropods that were included in the group as described in Le Quéré et al. (2016) can do this, but euphausiids do not.

The macrozooplankton group in Le Quéré et al. (2016) does not include salps and pteropods, it is the same group as used here, only including crustacean macrozooplankton. An excerpt from that paper describing the group is as follows "PlankTOM10 incorporates six autotrophic and four heterotrophic PFTs: [...] and crustacean macrozooplankton (euphausiids, amphipods, and others, called "macrozooplankton" for simplicity)". For the construction of the model, we first assign relative preferences among PFTs based on predator-prey ratio. For macrozooplankton, this preference is reduced for $N_2$-fixers, bacteria and picophytoplankton, as suggested by this reviewer. However, because the absolute value of the preference does not affect the PFT distribution, but only its biomass, we use the size of the preference to tune the model to observed biomass. This tuning method is explained in Le Quéré et al. (2016), Section 2.1 (p. 4115). To clarify this point, we added in our manuscript a sentence to explain this;

"Once the relative preference is established, the absolute value of the preference is tuned to improve the biomass of the different PFTs, as in Le Quéré et al. (2016)".

**2b. L103-104:** This is not true; many parameters have been modified. e.g. L260-266.

It was meant that the growth rate was the only change to the equations behind the parameter values. The following sentence has been modified to better describe this;

"The formulation of the growth rate is the only equation that has changed since the previous version of the model (Le Quéré et al., 2016), although many parameters have been modified (Section 2.1.6)."

**2c. L129-143:** Please mention here that the g Zj Fk term is a temperature-dependent Michaelis-Menten form that includes the prey preference and a half saturation coefficient. Otherwise, please provide the full equations.

Michaelis-Menten and grazing preference have been added in this section as suggested, along with a reference to section 2.1.2, which goes into further detail and now includes the full equation (see response to Referee comment 2f).

"For growth through grazing, $g_{F_k}^{Z_j}$ is the grazing rate by zooplankton $Z_j$ on food source $F_k$. This is a temperature-dependent Michaelis-Menten term that includes grazing preference (see section 2.1.2.). $MGE$ is the modelled growth efficiency."

**2d. Eq 1:** The authors mention in the Introduction that jellyfish are part of the biological pump and may be a significant vector for carbon export. How does the jellyfish PFT contribute to the 3 detrital pools? Are the parameters the same as or different than the other zooplankton groups?

To respond to this comment, a new section has been added to the Methods called 'Organic Carbon Cycling Through the Plankton Ecosystem', which describes the contribution of PFTs to each pool, along with a schematic (Figure 1b) of the processes in the organic carbon cycle in PlankTOM.

"**2.1.5. Organic Carbon Cycling Through the Plankton Ecosystem**
In PlankTOM11, the growth of phytoplankton modifies dissolved inorganic carbon into DOC, which then aggregates into $POC_S$ and $POC_L$. $POC_S$ is also generated from protozooplankton egestion and excretion and is consumed through grazing by all zooplankton. $POC_L$ is also generated by aggregation from $POC_S$, egestion and excretion by all zooplankton, and from the mortality of mesozooplankton, macrozooplankton and jellyfish, and is consumed through grazing by all zooplankton. The portion of $POC_S$ and $POC_L$ which is not grazed, sinks through the water column and is counted as export production at 100m (Figure 1a). The sinking speed of $POC_S$ is 3 m/d$^{-1}$ and the sinking speed of $POC_L$ varies, depending on the concentration of ballast and the resulting particle density. Proto-, meso- and macrozooplankton excretion is largely in the form of particulate and solid faecal pellets, while this makes up very little of jellyfish excretion. Jellyfish instead produce and slough off mucus as part of their feeding mechanism (Pitt et al., 2009), which is represented in the model in the same way as the faecal pellet excretion, as a fraction of unassimilated grazing contributing to $POC_L$."

**2e. L165-166:** What is the GGE value for the jellyfish?

We added the value and a further explanation of the data used as follows;

"GGE is the portion of grazing that is converted to biomass. This was previously collated by Moriarty (2009) from the literature for crustacean and gelatinous macrozooplankton for the development of PlankTOM10. We extracted the data for jellyfish from this collation (all scyphomedusae) which gave an average GGE of $0.29 \pm 0.27$, n=126 (Moriarty, 2009)."

**2f. L181-183** and Table 3: The preferences are a ratio of what to what? What are the numerator and denominator? It is unclear how preference enters into the grazing equation and how preferences can be greater than one. I had to refer back to the Le Quéré et al. 2016 paper (LQ16). I think that this information is essential and the reader should not have to look at the original paper.

The description of preferences has been expanded and the grazing equation added.

"The zooplankton relative preferences are based around a predator-prey size ratio, which by design is set to 1 for zooplankton-diatom. Preferences to other PFTs and to particular carbon are then set relative to the preference for diatoms. The preference ratios are weighted using the global carbon biomass for each type against a total food biomass weighted mean (sum of all the PFTs), calculated from the MAREDAT database, following the methodology used for the other PFTs (Buitenhuis et al., 2013; Le Quéré et al., 2016). Zooplankton grazing is calculated using:

$$g_{F_k}^{Z_j} = \mu^T \frac{p_{F_k}^{Z_j}}{K_{1/2}^{Z_j} + \sum p_{F_k}^{Z_j} F_k}$$

where $g_{F_k}^{Z_j}$ is the grazing rate by zooplankton $Z_j$ on food source $F_k$ as shown in Eq. 1, where $\mu^T$ is the growth rate of zooplankton (Eq. 3), $p_{F_k}^{Z_j}$ is the preference of the zooplankton for the food source (prey) and $K_{1/2}^{Z_j}$ is the half saturation constant of zooplankton grazing."

**2g. L259-260:** These new parameters should be given for transparency and reproducibility. A table in the Appendix would be acceptable.
A table has been added to the Appendix (Table A4) with the old and new parameter values.

**2h. L279-290:** This description lacks some necessary details. For example, it says that PlankTOM10 is the same as PlankTOM11 except that the top predator mortality terms for mesozooplankton and macrozooplankton were returned to pre-jellyfish values.
One, I assume that pre-jellyfish values are those from Le Quéré et al. 2016 (but Table 5 does not support this).
See comment below.
Two, did the mesozooplankton respiration rate also return to pre-jellyfish values? (Yes, as evidenced in Table 5, but not mentioned in the text)
Added to text, see paragraph below.
Three, all of the changes to phytoplankton and bacteria made to PlankTOM11 remained?
Yes, further clarification has been added, see paragraph below.
Does PlankTOM10.5 just have two identical macrozooplankton groups?
Yes, further clarification has been added as follows
"This is done by parameterising the jellyfish PFT identically to the macrozooplankton PFT, so that there are 11 PFTs active, with two identical macrozooplankton. This simulation is called PlankTOM10.5."
I would suggest a table that outlines the differences between all 3 simulations. The authors attempted to do this with Table 5, but it lacks a column for PlankTOM10.5 and there are discrepancies between the text here and the values in the table.
Table 5 has been updated with the column suggested and the apparent discrepancies have been clarified in the text, see paragraph below.
If PlankTOM10 mortality rates were returned to pre-jellyfish values, then why are there different MAC mortality rates for PlankTOM10 (LQ16) and PlankTOM10 (this study) in the table?
Further clarification has been added to the text as follows (also for comments above).
"The first simulation sets the jellyfish growth rate to 0, so that it replicates the model set up with 10 PFTs in Le Quéré et al. (2016), here called PlankTOM10$^{LQ16}$, but it includes the updated growth formulation (section 2.1.1) and additional tuning (section 2.1.5). The simulation is labelled 'PlankTOM10' in the figures. The simulation is otherwise identical to PlankTOM11 except for the mortality term for macrozooplankton and the respiration term for mesozooplankton, which were initially returned to PlankTOM10$^{LQ16}$ values, to account for the lack of predation by jellyfish. Macrozooplankton mortality was then tuned down from the PlankTOM10$^{LQ16}$ value, from 0.02 to 0.012, to account for the change to the growth calculation (Table 5)."
And are macrozooplankton mortality and mesozooplankton respiration the only 2 parameters that varied across the 3 simulations?
Other than jellyfish, yes. The number of other changes made was kept to a minimum so that differences could be readily attributed to the changes in jellyfish. An additional sentence has been added to explain this;
"Otherwise, these simulations were identical to PlankTOM11. The number of differences between simulations, outside of the jellyfish PFT, were intentionally kept to minimum to allow for differences in results to be directly attributed to the changes to the jellyfish PFT."

**2i. Table 4:** Please show all parameters for jellyfish and macrozooplankton. e.g. grazing rate temperature-dependence, half saturation coefficients, MGE, GGE, etc.

Table 4 has been extended per your recommendation and references to it have been added in the text where equations are given.

**2j. Table 6:** What is adapted from Le Quéré et al. 2016? Is the PlankTOM10 in this table PlankTOM10-LQ16 or PlankTOM10-here? That needs to be noted in the table caption.

PlankTOM10 is the simulation run for this study. The table structure is from Le Quéré et al. (2016), but is probably not necessary to reference here, as the simulation results are all from this paper. The reference has been removed to avoid confusion over PlankTOM10.

**2k. Figure 8:** The black boxes that denote the North, Tropic, and South regions used in Figures 4 and 9 are only in the Pacific. If this analysis was only carried out for the Pacific, then that should be mentioned in the main text as well.

Added in text where appropriate, as follows;
"PlankTOM11 also reproduces higher chlorophyll concentrations in the Northern Pacific than the Southern (Fig. 9)"
"PlankTOM11 closely replicates the chlorophyll ratio between the north and south Pacific with a ratio of 2.12, compared to the observed ratio of 2.16 (Fig. 9)."

**3.** The analysis is rather superficial. It only described changes in plankton biomasses. I expected Section 3.3 to also compare differences in net primary production, carbon export, nutrient cycling, etc. with and without jellyfish. More detail should be given on how rates and fluxes change across the three simulations as well as on the mechanisms and processes involved. e.g. What are the grazing mortality rates of each PFT by each predator with and without jellyfish? How does the productivity of each PFT change? What are the flows of mass/energy from one PFT to another in the three simulations? These types of analyses would better elucidate food web structure and function with and without jellyfish.

Primary production and carbon export results (Fig. 10 and Fig. A2), analysis (within Section 3.3 Role of Jellyfish in the Plankton Ecosystem) and discussion have been added.
For 'grazing mortality rates of each PFT by each predator' and 'flows of mass/energy from one PFT to another' please see the next comment.

**3a. L421-422**: Why merely suggest that the decrease in predation of protozooplankton by macrozooplankton may be compensated for by the increase in predation by jellyfish? Why not use the model output to verify this? This is one example of how this manuscript would be improved by a more thorough analysis. For example, the mortality of each PFT could be partitioned by each grazer in the 3 simulations.

We have introduced a new figure to the appendix (Figure A2) to compare the flow of carbon between PFTs and into the organic carbon pools and the grazing rates of PFTs, along with additional analysis to substantiate this claim, using the model output already available.

**3b. L389-395:** The partitioning of phytoplankton biomass by PFT differs between the PlankTOM11 and observations. Is the PlankTOM10 in Table 6 PlankTOM10- LQ16 or PlankTOM10-here? Was the partitioning of different phytoplankton PFTs in PlankTOM10-LQ16 the same as here with PlankTOM11? Or did the partitioning change? If it stayed the same, it suggests that the jellyfish had no effect on phytoplankton community composition. If it changed, did it become more or less aligned with observations and how did the jellyfish affect it?

Table 6 is the PlankTOM10-this study, the reference to Le Quéré et al (2016) has been removed from the table description (see earlier comment). The percentage share of the plankton type for phytoplankton has been edited; mixed-phytoplankton has been removed from the calculation to

align it with the observations where there is no mixed-phytoplankton data. So, the percentages are of total phytoplankton biomass minus mixed phytoplankton. This change has been added to the table description. The following text has been added to the paragraph in question;
"The phytoplankton community composition changed from PlankTOM10$^{LQ16}$ to PlankTOM11, with some phytoplankton types moving closer to observations and some moving further away. For example, for N$_2$-fixers PlankTOM11 is in line with the upper end of observations at 8%, while PlankTOM10 and PlankTOM10$^{LQ16}$ overestimate N$_2$-fixers (10% and 11% respectively). For picophytoplankton, PlankTOM10$^{LQ16}$ is within the range of observations at 38%, while PlankTOM11 and PlankTOM10 underestimate the community share of picophytoplankton (17% and 20% respectively). For *Phaeocystis*, all three simulations underestimate the community share, but PlankTOM11 and PlankTOM10 (both 22%) are closer to the lower end of observations (27%) than PlankTOM10$^{LQ16}$ (15%; Table 6; Le Quéré et al, 2016). Overall, the difference between PlankTOM10$^{LQ16}$ and PlankTOM11 is greater than the difference between PlankTOM10 and PlankTOM11, suggesting that the change to growth of PFT's had a larger effect on phytoplankton community composition than the addition of jellyfish. This is expected, as the growth change directly effects each PFT and model results are sensitive to PFT growth rates (Buitenhuis et al., 2006, 2010). Jellyfish affect phytoplankton community composition, but the effect is small."

**3c. L413-414:** It is very unclear why the biomass of macrozooplankton drops from PlankTOM10-here to PlankTOM10.5 and this needs to be described in further detail. As far as I can tell, PlankTOM10-here and PlankTOM10.5 are nearly identical, except that macrozooplankton mortality and mesozooplankton respiration are lower in PlankTOM10.5 and there are two identical macrozooplankton groups.
Yes, that is correct.
Without any parameter changes, I expect the sum of the two macrozooplankton groups in PlankTOM10.5 to equal the biomass of the macrozooplankton in PlankTOM10-here. But the decrease in macrozooplankton mortality would lead me to expect an increase in macrozooplankton biomass in PlankTOM10.5. Why does it decrease? Did the drop in mesozooplankton respiration allow them to outcompete macrozooplankton for shared resources?
The following text has been added to section 3.3;
"The addition of this 11th PFT at the same trophic level reduces the biomass of the macrozooplankton (Fig. 10 and Fig. 11), despite the macrozooplankton mortality being reduced from PlankTOM10 to PlankTOM10.5 (Table 5) which would be expected to increase macrozooplankton biomass. However, the low level of mutual predation between the two macrozooplankton PFTs slightly reduces their overall biomass."
And what accounts for the change in latitudinal distribution of the mesozooplankton and macrozooplankton in PlankTOM10.5?
The change in the latitudinal distribution of macrozooplankton, especially in temperate regions, is driven by the competition with, and predation by jellyfish. The description of this has been expanded as follows;
"The greatest difference in PFT biomass, especially macrozooplankton biomass, between simulations occurs in latitudes higher than 30º where jellyfish biomass is highest (Fig. 10). In the tropics, jellyfish have a low impact on the ecosystem due to their low biomass in this region (Fig. 6 and Fig. 10). "
…
"The drop in mesozooplankton respiration from PlankTOM10 to PlankTOM10.5 (Table 5) lowers the rate of respiration, especially at lower temperatures. This likely accounts for the increase in PlankTOM10.5 mesozooplankton biomass at the higher latitudes (Fig. 10)."

**3d.** The Discussion does a thorough job addressing the assumptions and limitations of the model. However, it is lacking a section that describes the hypothesized mechanisms involved in the differences between PlankTOM10-LQ16, PlankTOM10-here, PlankTOM10.5, and PlankTOM11.

The mechanisms involved in the differences between the PlankTOM simulations are based around increasing competition at the top of the modelled food web, with increasing complexity at the top of the food web driving greater seasonal variability down through the food web via trophic cascades, described at various points throughout the discussion and has been expanded on at various points of the discussion in relation to reviewer comments. Please see the full revised discussion which nearly doubled in length, and the detailed points throughout this response.

**3e.** The Discussion also lacks a section on how the lower temperature sensitivity of jellyfish (lower Q10) compared to macrozooplankton might affect spatial distributions and how this is balanced or offset by disparities in their respiration and mortality rates.

There is some discussion of this in the first paragraph of the discussion, on the interplay between the temperature sensitivity of growth and mortality for jellyfish and macrozooplankton. This has been expanded as follows;

"Model results suggest high competition between macrozooplankton (crustaceans) and jellyfish. The growth rate of jellyfish is higher than that of macrozooplankton for the majority of the ocean (where the temperature is less than ~25°C) but the mortality of jellyfish is also significantly higher than macrozooplankton, again for the majority of the ocean. The combination of high growth and mortality means that jellyfish have a high turnover rate in temperate waters. In situations where jellyfish mortality is reduced (but still higher than macrozooplankton mortality), jellyfish outcompete macrozooplankton for grazing. Below 20°C jellyfish and macrozooplankton respiration is almost the same, so will have minimal influence on their relative biomass. Biomass is not linearly related to the growth, respiration and mortality rates, with biomass also dependent on prey availability, total PFT biomass and other variables. Because jellyfish also prey directly on macrozooplankton, the biomass of macrozooplankton can rapidly decrease in a positive feedback mechanism. Within oligotrophic regions both jellyfish and macrozooplankton biomass is low, as expected due to limited nutrients limiting phytoplankton growth in these regions. Around equatorial upwelling regions, macrozooplankton outcompete jellyfish. Macrozooplankton also outcompete jellyfish in many coastal areas including around northern Eurasia because they have a built-in coastal and under-ice advantage to represent enhanced recruitment in these environments which likely tips the balance in their favour (Le Quéré et al., 2016). Around 40°S and 40-50°N jellyfish mostly outcompete macrozooplankton; water here is around 10-17°C which is a temperature were jellyfish growth is the most above macrozooplankton growth and macrozooplankton mortality nears jellyfish mortality, which combined favour jellyfish over macrozooplankton. This sensitivity of the composition of the zooplankton community to the mortality of jellyfish could help explain why jellyfish are seen as increasing globally. A reduction in jellyfish mortality during early life-stages i.e. through reduced predation on ephyrae and juveniles by fish (Duarte et al., 2013; Lucas et al., 2012), could quickly allow jellyfish to outcompete other zooplankton, especially macrozooplankton."

**3f. L509-511:** More detail should be added here to describe where regionally, when seasonally, and which phytoplankton PFTs.

The following text has been added to this paragraph, along with an additional figure (Figure A1);
"The top down trophic cascade from jellyfish on the other zooplankton also changes some of the grazing pressures on the phytoplankton, which translates into regional and seasonal effects on chlorophyll. Jellyfish increase chlorophyll in the northern pacific and reduce it in the southern pacific, relative to PlankTOM10 (Fig. 9). Seasonally, in the global north jellyfish increase phytoplankton biomass most during the summer and in the global south jellyfish decrease phytoplankton biomass most during the summer, relative to PlankTOM10 (Fig. 11). In the north, most of this summer increase in phytoplankton comes from coccolithophores and *Phaeocystis*, while in the south most of the summer decrease comes from coccolithophores, picophytoplankton and mixed-phytoplankton (Fig. A1)."

**4.** The model really only characterizes the pelagic phase of the complex jellyfish life cycle. The authors cited much variability in this life cycle, but do not provide enough information on how representative this model is without the life cycle and dependence on benthic substrate? Some useful details include how much time is spent the pelagic medusa stage and how much biomass is present in this stage in comparison to the other stages.

The paragraph on the exclusion of a jellyfish life cycle in PlankTOM11 in the discussion has been expanded to read as follows;

"A key limitation of the representation jellyfish in the model is the exclusion of the full life cycle. Most jellyfish display metagenesis, alternating between a polyp phase that reproduces asexually and a medusa phase tat reproduces sexually (Lucas and Dawson, 2014). PlankTOM11 currently only characterises the pelagic phase of the jellyfish life cycle, with parameters based on data from the medusae and ephyrae. The biomass of jellyfish is maximal during the pelagic medusa stage, as medusae are generally several orders of magnitude larger than polyps and one polyp can release multiple ephyrae into the water column (Lucas and Dawson, 2014). Although most hydromedusae persist in the plankton for short periods of time, larger scyphomedusae can live for 4-8 months and individuals in some populations can survive for more than a year by over wintering; something that may be facilitated by global climate change (Boero et al., 2016). Polyps develop from planula larvae within 5 weeks of settlement, and can persist far longer than medusae owing to their asexual mode of reproduction and the fact that they can encysts, which allows them to remain dormant until environmental conditions are favourable for budding (Lucas and Dawson, 2014). Unusually, mature medusae of *Turritopsis dohrnii* can revert back to the polyp stage and repeat the life cycle, which effectively confers immortality (Martell et al., 2016). Our understanding of polyp ecology is almost entirely based on laboratory reared specimens of common, eurytolerant species, with the patterns observed being locale- and species-dependent. We know that temperature changes can trigger the budding of ephyrae by scyphopolyps, which may lead to an increase in the medusa population (Han and Uye, 2010; Lucas and Dawson, 2014), but the number of species whose polyps have been located and studied in situ is minuscule and so estimates of polyp abundance or biomass are impossible even to estimate.

Models that include the full jellyfish life cycle are still relatively new, and their focus has been locale- and species-dependant (e.g. Henschke et al., 2018; Schnedler-Meyer et al., 2018). The aim of this study was not to reproduce small-scale blooms, but rather to assess at the large and global scale the influence of jellyfish on the plankton ecosystem and biogeochemistry. We consider it enough to note that higher temperature within PlankTOM11 increases the growth rate, which translates into increased biomass if there is sufficient food, thus providing a representation of an increasing medusa population. The inclusion of jellyfish life cycles into PlankTOM11 would introduce huge uncertainties due to the lack of clear in situ life cycle data and is beyond the scope of the exercise."

Other scientific questions/issues:

**L152-155**: Please add a statistical skill metric for the exponential fit and the 3-parameter fit to observations so that the reader may compare. Showing both in Figure 2 for the jellyfish could also help support the claim that the exponential fit is poor.

The line for exponential fit has been added to Figure 2, and Appendix Table A2 has been added giving the $R^2$ value for both fits and the number of observations per PFT.

**L228:** How was the adjusted mortality of $\mu = 0.12$ chosen from the sensitivity analysis? What skill metric was used?

The following explanation has been added to the text;

"This value was chosen based on expert judgement of the overall fit across multiple data streams. Whereas it was informed by the quantitative values in Table 6, the final choice required the balance

of positive and negative performance that required expert judgement rather than a statistical number."

**L251-254:** Jellyfish had a higher preference for protists than microzooplankton. Why were changes unnecessary for the protist parameters?

Here we assume the reviewer means that jellyfish have a higher preference for protists (protozooplankton) than for macrozooplankton – there is no PFT called microzooplankton in PlankTOM11 (this is a generalisation since the study of Le Quéré et al. 2016). A sentence communicating the reasoning behind no changes to the protozooplankton parameters has been added to this section. The order of paragraphs in this section has also been altered, to improve flow with the addition of this sentence;

"The jellyfish PFT is a significant grazer of macrozooplankton and mesozooplankton (Table 3), to account for this additional grazing the mortality term for macrozooplankton and the respiration term for mesozooplankton were reduced compared to model versions where no jellyfish are present (Table 5). Respiration is reduced in place of mortality for mesozooplankton as their mortality term had already been reduced to zero to account for predation by macrozooplankton (Le Quéré et al., 2016). The jellyfish PFT is also a significant grazer of protozooplankton, however, following the adjustment of protozooplankton grazing on picophytoplankton to account for changes to the growth rate formulation and the low sensitivity of protozooplankton to jellyfish mortality (Fig. 4) additional changes to protozooplankton parameters were found to be unnecessary."

**L272:** Why was 1948 not used for spin-up, since this would be the start year of meteorological forcing. Couldn't using 1980 to spin-up induce a shock to the system at 1948 that would then need to stabilize?

The use of 1980 for spin-up, rather than 1948, is because it represents a more 'average' year, as discussed in the paper, and has more weather data available. The following information has been added to the text to substantiate this decision;

"Year 1980 is used as a typical average year, as it has no strong El Nino/La Nina, as in Le Quéré et al. (2010). Furthermore, because of the greater availability of weather data (including by satellite) in 1980 compared to 1948, the dynamical fields are generally more representative of small-scale structures than the earlier years. There is a small shock to the system at the start of meteorological forcing, but this stabilises within a few years and decades before the model output is used for analysis. Tests of different spin-up years were carried out in Le Quéré et al. (2010), including both 1948 and 1980, with little impact on trends generally."

**L307-308:** Why were the MAREDAT observations binned to a different grid as the model? Why not use the same grid?

We used the same grid as provided for the other PFTs in the MAREDAT data so all PFTs were treated the same way. The published data were gridded to 1x1 (Buitenhuis et al., 2013), which is the grid used by the World Ocean Atlas.

**L338:** Please add a global map of observations for visual comparison.

A global map of JeDI observations (replicated from Lucas et al., 2014) has been added to Figure 6. The maps of jellyfish biomass from PlankTOM11 have been adapted to replicate the units and colour scale (log) in the Lucas panel. The following text has been added to section 3.1 to accompany the adapted figure 6;

"However, PlankTOM11 underestimates the range of observations in the top 200m (Fig. 6). PlankTOM11 overestimates the minimum values and underestimates the maximum values. However, part of this discrepancy may be due to under-sampling in the observations. For instance, the highest values in the observations (>100, orange and darker) are from cells with the lower number of observations (Fig 6; Lucas et al., 2014)."

**L348-349:** Why not use the same type of mean to compare? Or if the authors are concerned about the underrepresentation of zeros in the observations, why not use the mean that is best for describing that type of distribution?

The references for estimations of jellyfish biomass use different types of mean, so the authors felt it was best to compare using different types of mean as in the published references. The different types also give a better view of the spread of the data, as opposed to just one. E.g. MAREDAT papers provide the median and the average (described as the min and max) of plankton biomass to show the range within the observations.

**L352-361:** This paragraph is missing a sentence that notes where the model disagrees with observations spatially. There is a prominent difference out the outer shelf of the Eastern Bering Sea where the model predicts some of the highest biomasses while observations show some of the lowest biomasses. A potential explanation for this discrepancy should also be added here or in the Discussion as appropriate. (Here if it is local to Alaska, in the Discussion if it is applicable to the model as a whole.)

The following two sentences have been added;

"Spatially, the observations peak around the north coast of Alaska while PlankTOM11 peaks around the south coast (Fig. 7). This difference is likely due to the lack of small-scale physical processes in the model due to the relatively coarse model resolution."

**L376-379:** Is this ratio standard for validating model chlorophyll? i.e. Is it a meaningful metric?

Yes, for this model family the ratio is standard. A comparison to the ratio in PlankTOM10$^{LQ16}$ has been added, along with a justification for using the metric;

"PlankTOM11 closely replicates the chlorophyll ratio between the north and south Pacific with a ratio of 2.12, compared to the observed ratio of 2.16 (Fig. 9). This is an improvement on PlankTOM10$^{LQ16}$, which had a chlorophyll ratio of 1.72 (Le Quéré et al., 2016). PlankTOM10 and PlankTOM10.5 underestimate the observed ratio with ratios of 1.57 and 1.96 respectively (Fig. 9). …

The north/south chlorophyll ratio metric was developed by Le Quéré et al. (2016) as a simple method to quantify model performance for emergent properties, focussing on the Pacific Ocean as the area where this ratio is most pronounced in the observations. These simulations further support the suggestion by Le Quéré et al. (2016) that the observed distribution of chlorophyll in the north and south is a consequence of trophic balances between the PFTs and improves with increasing plankton complexity."

**L387:** This underestimate of primary production by 10 PgC/y seems rather large. How does it compare to the Le Quéré et al. 2016 model and other biogeochemical models?

A comparison to Le Quéré et al 2016 has been added, along with a possible explanation for the consistent underestimation;

"PlankTOM11 underestimates primary production by 10 PgC y$^{-1}$, which is similar to the underestimation in PlankTOM10$^{LQ16}$ of 9 PgC y$^{-1}$. As suggested by Le Quéré et al. (2016) this may be due to the model only representing highly active bacteria, which is unchanged between the model versions, while observed biomass is also from low activity bacteria and ghost cells."

**L395-398:** These statements could be supported by mentioning that the light affinity and nutrient uptake parameters of mixed phytoplankton and Coccolithophores are very similar to those of picophytoplankton, with the exception of Fe uptake.

Thank you for the suggestion, this has been included;

"The observations are dominated by picophytoplankton, followed by *Phaeocystis* and Diatoms (Table 6). Mixed-phytoplankton and Coccolithophore parameters for light affinity and nutrient uptake are similar to those of picophytoplankton, with the exception of iron uptake. The modelled

mixed-phytoplankton is likely taking up the ecosystem niche of picophytoplankton. Coccolithophores are overestimated by a factor of 10 and may also be filling the ecosystem niche of picophytoplankton in the model (Table 6)."

**L406-407:** But jellyfish have a much higher preference for mesozooplankton and protozooplankton than macrozooplankton. How does this affect the results?
The following text has been added to this section;
"Macrozooplankton exhibit the largest change in biomass between the three simulations, followed by mesozooplankton (Fig. 10). This is despite the higher preference of jellyfish grazing on mesozooplankton (ratio of 10) than on macrozooplankton (ratio of 5; Table 3). The central competition for resources between jellyfish and macrozooplankton is that they both preferentially graze on mesozooplankton, then on protozooplankton, although macrozooplankton have a lower preference ratio for zooplankton than jellyfish, as more of their diet is made up by phytoplankton (Table 3). In simple terms this means that for two equally sized populations of jellyfish and macrozooplankton, jellyfish would consume more meso- and protozooplankton than would be consumed by macrozooplankton. However, predator biomass, prey biomass and the temperature dependence of grazing interact to affect the rate of consumption (Eq. 5)."

**L432-433:** This line is too vague. What was the largest direct influence of jellyfish? Predation? Competition? If competition, for which resources?
The following text has been added;
"The largest direct influence of jellyfish in these regions is its control on macrozooplankton biomass, through competition for prey resources, particularly mesozooplankton and protozooplankton, and through the predation of jellyfish on macrozooplankton."

**L434-437:** Is a double peak in northern hemisphere phytoplankton seasonal biomass consistent with observations? Is the amplification more or less similar to observations? Is one of the simulations (PlankTOM10-LQ16, PlankTOM10-here, PlankTOM10.5, PlankTOM11) more similar to observations?
The following text and analysis have been added to this section, along with Table A6;
"Observations (MAREDAT) show two peaks in phytoplankton biomass although the peaks are offset in timing from all three PlankTOM simulations. The amplitude of the full seasonal cycle in observations is $0.78 – 2.67$ µmol C/L (median – mean) with all three PlankTOM simulations falling well within this range (Table A6). Removing the winter months, where there is less variability, gives a non-winter observational amplitude of $0.7 – 2.12$ µmol C/L. PlankTOM11 is the highest, with a non-winter amplitude of $0.97$ µmol C/L, with the other two simulations lower at $0.8$ µmol C/L (PLankTOM10.5) and $0.81$ µmol C/L (PlankTOM10; Table A6). PlankTOM10$^{LQ16}$ has a lower seasonal amplitude than PlankTOM11, although a slighter higher non-winter amplitude by $0.05$ µmol C/L (Table A6)."

**L486-488:** The jellyfish may not need a coastal advantage, but a deep-water disadvantage, since their benthic polyps are filter feeders and dependent on pelagic plankton.
Thank you for this suggestion, it has been added to the end of this paragraph as a potential avenue for future research.
"Alternatively, a deep-water disadvantage could be introduced for jellyfish to introduce an element of their life cycle dependencies in that the polyps require benthic substrate for settlement and development into the next life stage and are dependent on plankton for food, which are more abundant in shallower coastal waters. Future work on PlankTOM11 could investigate the strengths and weaknesses of these two avenues (coastal advantage and deep-water disadvantage) for introducing a jellyfish lifecycle element to the model."

**Figure 2**: Using the same y-axis scale for all subplots hides the fit with observations for FIX, MAC, and JEL. Also, an R2 and/or p-value for the fit would be appreciated.

The y-axis has been adjusted for FIX, MAC and JEL. A table has been added to the appendix with the $R^2$ for the old two-parameter fit and the new 3-parameter fit (Table A2).

**Figure 4:** Could observations of the observed PFT biomasses from MAREDAT be added to this plot similar to observed chlorophyll?

The range of the observed PFT biomasses (given in Table 6) are much larger than these plots, so including them hides the variability of the sensitivity tests, which is the key message in this Figure.

**Figures 5, 6, 7, 8:** I recommend a colormap that is perceptually ordered for spatial distributions. See cmocean, colorbrewer, and colormoves for examples.

These figures have been redrawn with a cmocean palette, with the exception of figure 6 which has been redrawn to replicate the colormap in Lucas et al. (2014).

**Figure 7:** The cyan color used in the time series is very difficult to see. Use a darker color or a dashed black line.

The lines have been changed to a darker blue.

**Figures 10, 11**: I would not refer to the 2nd macrozooplankton group of PlankTOM10.5 as jellyfish in these figures. Instead, the biomass of the 2 macrozooplankton groups should be summed together and displayed that way. Keeping them distinct is misrepresentative.

The second MAC (previously JEL) in PlankTOM10.5 has been summed with the other MAC PFT, with the figures redrawn and captions updated with the following addition;

"For PlankTOM10.5 the MAC PFT has been summed with the 11[th] PFT that duplicates MAC."

---

## Author Comment (AC2) · 7 Oct 2020

We sincerely thank the reviewer for their insightful comments, which have greatly helped to improve and clarify our manuscript. We respond in detail to each comment in the attached document, and clarified the points raised by the reviewer. In response to these comments we have implemented the following general changes:

- All parameter values for the jellyfish PFT have been added, with the expansion of tables in the main manuscript and the addition of tables in the appendices, along with further descriptions of the model structure and additional equations.
- Carbon and primary production have been added to the results section and discussion, along with diagrams of the carbon flow within the model.
- The discussion has been expanded throughout to about twice its original length.

We provide full details in the attached, with our replies to reviewers' comments in blue, and the new text provided in the manuscript highlighted in red.

Rebecca Wright, on behalf of the author team.

**Response to the comments**

**Reviewer #2**

General Comments

The manuscript fits perfectly in the scope of Biogeosciences in that it considers the role of plankton within marine biogeochemistry. It uses a currently developed model PlankTOM10 and adds a jellyfish plankton functional traits (PFT) to further resolve the global ocean plankton system. Gelatinous zooplankton within the Cnidaria and Ctenophora have been neglected in virtually all models, yet we know that they have the potential to play a significant role in structuring plankton food webs directly and indirectly via predation, and facilitate the flux of organic carbon to the seafloor via the production of mucous, messy feeding, carcasses (also known as jelly falls) and to a much lesser extent faecal pellets. The pelagic tunicates (salps, doliolids) can form substantive bloom events and have the capacity to graze particles down to only a few microns, but these have also not been included. Although the authors appear to have used appropriate methods and scientific assumptions, it is difficult to make firm judgements about this as there not a detailed justification of the parameter values used. I would expect in a paper such as this for a detailed summary of all sources of parameter inputs listed in the Appendix for readers to check themselves and also reproduce the models. Without this a reader has to go on a fact finding mission themselves. The interpretation of the results are sound but not particularly substantive. The inclusion of jellyfish in the PlankTOM model is a significant step forward but the authors have not really explored this as much as I would have hoped. When I first read the title of the paper I expected a model of carbon flows from one PFT to another presented rather than just biomass outputs, and this left me feeling somewhat disappointed. If these concerns were addressed it would make the work far more powerful and novel.

Thank you for your comments. All new and altered parameter values have been added, either within the main manuscript or in the appendix. Carbon flow results, analysis and discussion have also been

included, with the discussion section expanded to almost double its previous length. Please see below for specific details.

Specific Comments

**L33** - I am not sure jellyfish play a unique role. They do play a role, as do all the other functional groups. What are you suggesting by the term unique?
'Unique' had been removed from the title and abstract. Although all PFTs are unique, indeed there is maybe not a case here to argue why jellyfish are more unique than others.

**L34** - There have been a (very few) instances where jellyfish have been considered in plankton ecosystems, e.g., Ruzicka et al 2012 for the California Current System.
Previous examples of jellyfish modelling have been added to the discussion, the sentence here has been adapted to highlight that there has been no previous inclusion of jellyfish in a **global** plankton ecosystem model.
In the abstract;
"Overall the results suggest that jellyfish play an important role in regulating global marine plankton ecosystems, which has been generally neglected so far."
In the discussion;
"Jellyfish have been included in a range of regional models, the majority are fisheries-based ecosystem models, namely ECOPATH and ECOPATH with ECOSIM (Pauly et al., 2009). These include regional models of the Northern Humboldt Current system (Chiaverano et al., 2018), the Benguela Upwelling System (Roux et al., 2013; Roux and Shannon, 2004; Shannon et al., 2009) and an end-to-end model of the Northern California Current system, based on ECOPATH (Ruzicka et al., 2012). Jellyfish have also been included in regional Nutrient Phytoplankton Zooplankton Detritus (NPZD) models, representing small-scale coastal temperate ecosystems with simple communities, for example, Schnedler-Meyer et al. (2018) and Ramirez-Romero et al. (2018). These models have provided valuable insight into jellyfish in the regions studied, but the focus on coastal ecosystems and either a top-down approach (ECOPATH) or a highly simplified ecosystem (NPZD) limits their scope. PlankTOM11 offers the first insight into jellyfish on a global scale, from a modelling perspective."

**L86** - You do not mention, or make clear, what the composition of the macrozooplankton group is. Does it include pelagic tunicates, which are going to graze down to a smaller food size than crustacean macrozooplankton such as euphausiids. This would make a difference to how your model runs.
In the following sentence macrozooplankton are described as 'crustaceans' and the reader is directed to Table 1 which contains definitions of the PFTs, where macrozooplankton is further described as 'euphausiids, amphipods and others'. We have added 'crustacean' to Table 1 and 'euphausiids' to the text descriptions to enhance clarification.

**p114** - What are the size definitions of the two particulate detrital pools? The terms small and large are vague. You do not specify the contribution of each of the PFTs to each pool. For example, jellyfish produce virtually nothing in the way of particulate / solid faecal pellets.
A new section has been added to the Methods 'Organic Carbon Cycling Through the Plankton' which describes the contribution of PFTs to each pool, along with a schematic (Figure 1b) of the processes in the organic carbon cycle in PlankTOM.
"**2.1.5. Organic Carbon Cycling Through the Plankton**
In PlankTOM11, the growth of phytoplankton modifies dissolved inorganic carbon into DOC, which then aggregates into $POC_S$ and $POC_L$ (Fig. 1b). $POC_S$ is also generated from protozooplankton egestion and excretion and is consumed through grazing by all zooplankton. $POC_L$ is also generated

by aggregation from $POC_S$, egestion and excretion by all zooplankton, and from the mortality of mesozooplankton, macrozooplankton and jellyfish, and is consumed through grazing by all zooplankton. The portion of $POC_S$ and $POC_L$ which is not grazed, sinks through the water column and is counted as export production at 100m (Fig. 1b). The sinking speed of $POC_S$ is 3 m/d$^{-1}$ and the sinking speed of $POC_L$ varies, depending on particle and water density. Proto-, meso- and macrozooplankton excretion is largely in the form of particulate and solid faecal pellets, while this makes up very little of jellyfish excretion. Jellyfish instead produce and slough off mucus as part of their feeding mechanism (Pitt et al., 2009), which is represented in the model in the same way as the faecal pellet excretion, as a fraction of unassimilated grazing contributing to $POC_L$."

**p135** - What is MGE?
Modelled Growth Efficiency
"and $MGE$ is the modelled growth efficiency."

**p147** - You mention that jellyfish growth rates were compiled as a function of temperature from the literature, but you do not provide any indication of which papers were used, how many were used, which taxa the growth rates were compiled from etc. It is this level of detail that is absent from the methodology which makes traceability of the data impossible to verify.
A table of references for the growth values used has been added to the appendix (Table A1) and referenced in the text. The table also includes information on the species, life stage and number of data for each reference.

**p165-166** - Continuing with the issues of transparency, the values for GGE are obtained from the literature (Moriarty, 2009), but this is difficult to verify a that is a PhD thesis. You should make it clear that the data from the literature have been collated by Moriarty, 2009. How many values were collated? What are the range of values? Stating these will make readers far more confident about the inputs into the model.
"GGE is the portion of grazing that is converted to biomass. This was previously collated by Moriarty (2009) from the literature for crustacean and gelatinous macrozooplankton for the development of PlankTOM10. We extracted the data for jellyfish from this collation (all scyphomedusae) which gave an average GGE of $0.29 \pm 0.27$, n=126."

**p178** - Do you have evidence that ephyrae do have a higher clearance rate for autotrophs. There are not many papers that have analysed diet of ephyrae and there are mixed messages about diet. For example, how can you take into account selective vs non-selective feeders and time of year (relative to the spring phytoplankton blooms)?
The time of year (and spring phytoplankton blooms etc) occur as emergent properties within the model as the PFTs react to temperature and light changes, rather than being directly accounted for in the preferences or parameterization. Selective vs non-selective feeders are not accounted for, grazing depends on the biomass of each PFT in that location and the temperature. The sentences on ephyrae feeding has been edited for clarity as follows;
"There is little evidence in the literature for jellyfish actively consuming autotrophs. One of the few pieces of evidence is a gut content analysis where 'unidentified protists... some chlorophyll bearing' were found in a small medusa species (Colin et al., 2005). Another is a study by Boero et al. (2007) which showed that very small medusae such as Obelia will consume bacteria and may consume phytoplankton. Studies on the diet of the ephyrae life cycle stage are limited in comparison to those on medusa, but the literature does show evidence for ephyrae consuming protists and phytoplankton (Båmstedt et al., 2001; Morais et al., 2015). We assume that ephyrae are likely to have a higher clearance rate of autotrophs, due to their smaller size as with the small medusa, but this will have a minimal effect on the overall preferences and the biomass consumed, so preferences for autotrophs are kept low."

**L299** - It would be useful to include a map showing the global distribution of jellyfish for the reader to gain a better understanding of the spatial distribution and coverage.

A global map of JeDI observations (replicated from Lucas et al., 2014) has been added to Figure 6. The maps of jellyfish biomass from PlankTOM11 have been adapted to replicate the units and colour scale (log) in the Lucas panel. The following text has been added to section 3.1 to accompany the adapted figure 6;

"However, PlankTOM11 underestimates the range of observations in the top 200m (Fig. 6). PlankTOM11 overestimates the minimum values and underestimates the maximum values. However, part of this discrepancy may be due to under-sampling in the observations. For instance, the highest values in the observations (>100, orange and darker) are from cells with the lower number of observations (Fig 6; Lucas et al., 2014)."

Are the Cnidaria data used from the upper 200m only, as you indicate that in the original dataset jellyfish were available for a much wider depth range. Again, this is for transparency purposes.

The Cnidaria data for the whole depth range are used, this has been clarified in the text by adding the following sentence;

"Data from all depths are included in the analysis."

**L319 and 327** - Why do you express the values for jellyfish biomass as 0.46 - 3.11 pg C on line 319 in the methods and just 0.46 pg C on line 327 at the start of the results (where other published values are expressed as a range)?

This has been rectified so that the PlankTOM11 jellyfish biomass is always expressed as a range.

**L342-344** - It is obvious that the majority of jellyfish biomass is distributed around the coasts because a) that is where the majority of sampling has taken place and so there will be sampling bias, and b) it is likely that the majority of jellyfish collected are scyphomedusae with a metagenic life cycle requiring a hard substrate for the benthic polyp population.

The following sentences have been added to the end of this paragraph;

"A key caveat in jellyfish data is that the data is not uniformly distributed spatially or temporally and not proportionally distributed between various biomes of the ocean, with collection efforts skewed to coastal regions and the Northern Hemisphere (MAREDAT; Lilley et al., 2011; Lucas et al., 2014). This sampling bias and sampling methods also tend to favour larger, less delicate species, which are often scyphomedusae with a meroplanktonic life cycle."

**L417-418** - You state there is a high preference for jellyfish on protozooplankton. The vast majority of diet and feeding studies on jellyfish suggest that mesozooplankton are the preferred prey for the majority of jellyfish. Smaller taxa and juvenile forms (ephyrae) would consume protozooplankton, but this is not the case for most of the scyphomedusae. In the jellyfish dataset used for the PlankTOM11 model, are the classes or genera listed? If so, it would be helpful to briefly indicate the make up of the jellyfish community used in this study.

Further clarification of prey preferences has been added to the methods (see comment **p178**). A table has also been added to the Appendix (Table A3) with the references for jellyfish grazing, along with information on the jellyfish species and the prey preference for each reference.

The high prey preference of jellyfish for protozooplankton is in comparison to the PFT's other than mesozooplankton, which has the highest prey preference of any of the PFT's. The section has been edited as follows to clarify this and to also discuss the impact on mesozooplankton;

"The addition of jellyfish changes the zooplankton with the highest biomass from macrozooplankton to protozooplankton and reduces the biomass of mesozooplankton, in both the north and south (Fig. 11). However, the impact on the biomass of mesozooplankton and protozooplankton is small, despite mesozooplankton being the preferential prey of jellyfish, followed by protozooplankton. The small impact of jellyfish on mesozooplankton and protozooplankton biomass may be due to trophic

cascade effects where jellyfish reduce the biomass of macrozooplankton, which reduces the predation pressure of macrozooplankton on meso- and protozooplankton, whilst jellyfish simultaneously provide an additional predation pressure on meso- and protozooplankton. The decrease in predation by macrozooplankton may be compensated for by the increase in predation by jellyfish, resulting in only a small change to the overall biomass of mesozooplankton and protozooplankton."

**L453** - The grid resolution is stated as ~2o x 1o, but the original dataset were gridded in 1 x 1 degree. Why was the resolution changed?
We used the same grid as provided for the other PFTs in the MAREDAT data so all PFTs were treated the same way. The published data were gridded to 1x1 (Buitenhuis et al., 2013), which is the grid used by the World Ocean Atlas.

**L472** - Brotz et al. 2012 is not the most appropriate reference to support the description of jellyfish reproduction alternating between a sexually-reproducing pelagic medusa and asexually-reproducing benthic polyp, as the Brotz paper is about global distributions of gelatinous zooplankton and not reproduction.
The reference has been changed to Lucas and Dawson (2014).

**L472-479** - Be careful about saying that increasing temperatures increase growth of jellyfish, which they do, but it is an oversimplification of the whole life cycle as ephyrae are typically produced following colder than average winter temperatures (certainly for temperate populations of the common jellyfish Aurelia, which likely dominates datasets).
This whole paragraph on life cycles has been expanded and rewritten as follows;
"A key limitation of the representation jellyfish in the model is the exclusion of the full life cycle. Most jellyfish display metagenesis, alternating between a polyp phase that reproduces asexually and a medusa phase tat reproduces sexually (Lucas and Dawson, 2014). PlankTOM11 currently only characterises the pelagic phase of the jellyfish life cycle, with parameters based on data from the medusae and ephyrae. The biomass of jellyfish is maximal during the pelagic medusa stage, as medusae are generally several orders of magnitude larger than polyps and one polyp can release multiple ephyrae into the water column (Lucas and Dawson, 2014). Although most hydromedusae persist in the plankton for short periods of time, larger scyphomedusae can live for 4-8 months and individuals in some populations can survive for more than a year by over wintering; something that may be facilitated by global climate change (Boero et al., 2016). Polyps develop from planula larvae within 5 weeks of settlement, and can persist far longer than medusae owing to their asexual mode of reproduction and the fact that they can encysts, which allows them to remain dormant until environmental conditions are favourable for budding (Lucas and Dawson, 2014). Unusually, mature medusae of *Turritopsis dohrnii* can revert back to the polyp stage and repeat the life cycle, which effectively confers immortality (Martell et al., 2016). Our understanding of polyp ecology is almost entirely based on laboratory reared specimens of common, eurytolerant species, with the patterns observed being locale- and species-dependent. We know that temperature changes can trigger the budding of ephyrae by scyphopolyps, which may lead to an increase in the medusa population (Han and Uye, 2010; Lucas and Dawson, 2014), but the number of species whose polyps have been located and studied in situ is minuscule and so estimates of polyp abundance or biomass are impossible even to estimate."

Overall the discussion is rather brief and does not fully explore the differences between the different model outputs and the mechanisms driving those differences. The discussion feels rather superficial and far more explanation is needed to make it robust.
A substantial section on carbon fluxes in the model has been added to the discussion, and the original discussion has been expanded on throughout as per reviewer comments.

Technical Corrections

**L55** - benthic polyp (delete s) corrected

**L177** - Obelia in italics corrected

**L195** - Aurelia in italics corrected

**L197** - data were (not was) corrected

**L394** - picophytoplankton (lower case p) corrected

**Table 1** - italicise genus names corrected

---

## Author Comment (AC3) · 7 Oct 2020

[revised manuscript text omitted]

**Figure 6.** Jellyfish carbon biomass (mg C m³) in PlankTOM11 and the observed jellyfish carbon biomass (mg C m³) from JeDI (panel reproduced from Lucas et al., 2014) on a logarithmic scale. PlankTOM11 results (left) are the mapped monthly minimum, mean and maximum biomass from monthly climatologies. Observations (right) are the mean biomass (with no observations in pale blue). All data is for 0-200m. Only the mean values are available for the observations (Lucas et al. 2014), while for the model the min, mean and max are given. 
[revised manuscript text omitted]

[Figure]

**(a) PlankTOM11**

Phytoplankton
2.9 PgC

41.5 PgC/y

14.13 PgC/y

PRO
0.43 PgC

6.36 PgC/y

MES
0.33 PgC

5.51 PgC/y

MAC
0.29 PgC

3.51 PgC/y

JEL
0.26 PgC

BAC
0.6 PgC

DOC
31.68 PgC

POC$_S$
1.32 PgC

POC$_L$
0.76 PgC

7.11 PgC/y

→ mortality
→ primary production
→ egestion & excretion
→ deposition (river, dust & air)
→ aggregation
→ grazing
→ Sinking/Export

899

[Figure]

**(b) PlankTOM10/10.5**

Phytoplankton
2.85 PgC$^{T10.5}$
2.89 PgC$^{T10}$

38.18 PgC/y$^{T10.5}$
43.43 PgC/y$^{T10}$

12.00 PgC/y$^{T10.5}$
12.04 PgC/y$^{T10}$

PRO
0.40 PgC$^{T10.5}$
0.45 PgC$^{T10}$

5.95 PgC/y$^{T10.5}$
8.00 PgC/y$^{T10}$

MES
0.44 PgC$^{T10.5}$
0.34 PgC$^{T10}$

10.57 PgC/y$^{T10.5}$
10.33 PgC/y$^{T10}$

MAC
0.65 PgC$^{T10.5}$
0.46 PgC$^{T10}$

BAC
0.6 PgC$^{T10.5}$
0.6 PgC$^{T10}$

DOC
31.00 PgC$^{T10.5}$
39.81 PgC$^{T10}$

POC$_S$
1.31 PgC$^{T10.5}$
1.18 PgC$^{T10}$

POC$_L$
0.71 PgC$^{T10.5}$
0.74 PgC$^{T10}$

6.62 PgC/y$^{T10.5}$
6.96 PgC/y$^{T10}$

900

901 **Figure A2.** Schematic representation of global carbon biomass and rates in the PlankTOM marine
902 ecosystem model including sources and sinks for dissolved organic carbon (DOC) and small ($POC_S$)
903 and large ($POC_L$) particulate organic carbon. (a) PlankTOM11 and (b) PlankTOM10 and
904 PlankTOM10.5. Carbon biomass (PgC) of PFT's and organic carbon pools are given within boxes and
905 ovals, carbon rates (PgC/y) of primary production (light green), grazing (dark green) and export
906 production (purple) are given next to the corresponding arrows. All data are averaged for 1985 to
907 2015.

908

909

910